# Revisit the Power of Vanilla Knowledge Distillation: from Small Scale to Large Scale

**Zhiwei Hao**[1,2*]**, Jianyuan Guo**[3*]**, Kai Han**[2]**, Han Hu**[1†]**, Chang Xu**[3]**, Yunhe Wang**[2†]

[1]School of information and Electronics, Beijing Institute of Technology.
[2]Huawei Noah's Ark Lab.
[3]School of Computer Science, Faculty of Engineering, The University of Sydney.

{haozhw, hhu}@bit.edu.cn, jguo5172@uni.sydney.edu.au,
{kai.han, yunhe.wang}@huawei.com, c.xu@sydney.edu.au

## Abstract

The tremendous success of large models trained on extensive datasets demonstrates that scale is a key ingredient in achieving superior results. Therefore, the reflection on the rationality of designing knowledge distillation (KD) approaches for limited-capacity architectures solely based on small-scale datasets is now deemed imperative. In this paper, we identify the *small data pitfall* that presents in previous KD methods, which results in the underestimation of the power of vanilla KD framework on large-scale datasets such as ImageNet-1K. Specifically, we show that employing stronger data augmentation techniques and using larger datasets can directly decrease the gap between vanilla KD and other meticulously designed KD variants. This highlights the necessity of designing and evaluating KD approaches in the context of practical scenarios, casting off the limitations of small-scale datasets. Our investigation of the vanilla KD and its variants in more complex schemes, including stronger training strategies and different model capacities, demonstrates that vanilla KD is elegantly simple but astonishingly effective in large-scale scenarios. Without bells and whistles, we obtain state-of-the-art ResNet-50, ViT-S, and ConvNeXtV2-T models for ImageNet, which achieve 83.1%, 84.3%, and 85.0% top-1 accuracy, respectively. PyTorch code and checkpoints can be found at https://github.com/Hao840/vanillaKD.

## 1 Introduction

In recent years, deep learning has made remarkable progress in computer vision, with models continually increasing in capability and capacity [1, 2, 3, 4, 5]. The philosophy behind prevailing approach to achieving better performance has been *larger is better*, as evidenced by the success of increasingly deep [2, 6, 7, 8] and wide [9, 10, 11] models. Unfortunately, these cumbersome models with numerous parameters are difficult to be deployed on edge devices with limited computation resources [12, 13, 14], such as cell phones and autonomous vehicles. To overcome this challenge, knowledge distillation (KD) [15] and its variants [16, 17, 18, 19, 20, 21, 22, 23, 24] have been proposed to improve the performance of compact student models by transferring knowledge from larger teacher models during training.

Thus far, most existing KD approaches in the literature are tailored for small-scale benchmarks (*e.g.*, CIFAR [25]) and small teacher-student pairs (*e.g.*, Res34-Res18 [2] and WRN40-WRN16 [10]). However, downstream vision tasks [26, 27, 28, 29] actually require the backbone models to be

---

[*]Equal contribution. [†]Corresponding author.

37th Conference on Neural Information Processing Systems (NeurIPS 2023).

pre-trained on large-scale datasets (*e.g.*, ImageNet [30]) to achieve state-of-the-art performances. Only exploring KD approaches on small-scale datasets may fall short in providing a comprehensive understanding in practical scenarios. Given the availability of large-scale benchmarks and the capacity of large models [31, 3, 32, 33, 34], it remains uncertain *whether previous approaches will remain effective* in more complex schemes involving stronger training recipes, different model capacities, and larger data scales.

In this paper, we delve deep into this question and thoroughly study the crucial factors in deciding distillation performances. We point out the ***small data pitfall*** in current knowledge distillation literature: once a sufficient quantity of training data is reached, different conclusions emerge. For example, when evaluated on CIFAR-100 (50K training images), KD methods meticulously designed on such datasets can easily surpass vanilla KD. However, when evaluated on datasets with larger scale, i.e., ImageNet-1K (1M training images), vanilla KD achieves on par or even better results compared to other methods.

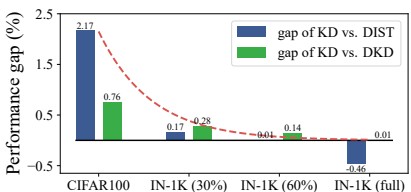

Figure 1: The performance gap between vanilla KD and other carefully designed approaches gradually diminishes with the increasing scale of datasets.

To break down this problem, we start by compensating for the limited data via training for longer iterations [35] on small-scale datasets. Albeit with longer schedule, meticulously designed KD methods still outperform vanilla KD by a large margin. It follows that large-scale datasets are necessary for vanilla KD to reach its peak.

We further investigate the crucial factors in deciding KD performances and carefully study two key elements, *i.e.*, training strategy and model capacity. In terms of different training strategies, we have made the following observations: (i) By evaluating the meticulously designed methods [18, 19] which perform well on small-scale benchmarks under different training recipes [16, 36, 37] on large-scale datasets such as ImageNet [30], it is evident that with the enhancement of data augmentation techniques and longer training iterations, the gap between vanilla KD [15] and other carefully designed KD methods gradually diminishes. (ii) Our experiments also demonstrate that logits-based methods [15, 18, 19] outperform hint-based methods [16, 17, 22, 21, 38] in terms of generalizability. With the growing magnitude of datasets and models, teacher and student models possess varying capacities to handle intricate distributions. Consequently, the utilization of the hint-based approach, wherein the student imitates the intermediate features of the teacher, becomes less conducive to achieving satisfactory results.

With regard to model capacity, we compare teacher-student pairs of different scales, *e.g.*, using Res34 to teach Res18 *vs.* using Res152 to teach Res50. The outcome reveals that vanilla KD ends up achieving on par performance with the best carefully designed methods, indicating that the impact of model capacity on the effectiveness of knowledge distillation is in fact small.

Throughout our study, we emphasize that due to ***small data pitfall***, the power of vanilla KD is significantly underestimated. In the meantime, meticulously designed distillation methods may instead become suboptimal when given stronger training strategies [36, 35] and larger datasets [39]. Furthermore, directly integrating vanilla KD to ResNet-50 [2], ViT-Tiny, ViT-Small [3], and ConvNeXtV2 [34] architectures leads to an accuracy of 83.1%, 78.1%, 84.3%, and 85.0% on ImageNet respectively, surpassing the best results reported in the literature [35] without additional bells and whistles. Our results provide valid evidence for the potential and power of vanilla KD and indicate its practical value. In addition, it bears further reflection on whether it is appropriate to design and evaluate KD approaches on small-scale datasets. Finally, we demonstrate that improving the backbone architecture with higher performance on ImageNet can also significantly enhance downstream tasks such as object detection and instance segmentation [26, 40, 41].

Although this work does not propose a novel distillation method, we believe our identification of ***small data pitfall*** and a series of analyzes based on this would provide valuable insights for the vision community in the field of knowledge distillation and the pursuit of new state-of-the-art results under more practical scenarios. Moreover, we anticipate that the released checkpoints of commonly used architectures will facilitate further research on downstream tasks.

## 2 Small data pitfall: limited performance of vanilla KD on small-scale dataset

### 2.1 Review of knowledge distillation

Knowledge distillation (KD) techniques can be broadly categorized into two types based on the source of information used by the pre-trained teacher model: those that utilize the teacher's output probabilities (logits-based) [19, 18, 15] and those that utilize the teacher's intermediate representations (hint-based) [16, 17, 22, 21, 42]. Logits-based approaches leverage teacher outputs as auxiliary signals to train a smaller model, known as the student:

$$\mathcal{L}_{\text{kd}} = \alpha \mathcal{D}_{\text{cls}}(\boldsymbol{p}^s, y) + (1 - \alpha)\mathcal{D}_{\text{kd}}(\boldsymbol{p}^s, \boldsymbol{p}^t), \tag{1}$$

where $\boldsymbol{p}^s$ and $\boldsymbol{p}^t$ are logits of the student and the teacher. $y$ is the one-hot ground truth label. $\mathcal{D}_{\text{cls}}$ and $\mathcal{D}_{\text{kd}}$ are classification loss and distillation loss, such as cross-entropy and KL divergence, respectively. The hyper-parameter $\alpha$ determines the balance between two loss terms. For convenience, we set $\alpha$ to 0.5 in all subsequent experiments.

Besides the logits, intermediate hints (features) [16] can also be used for KD. Considering a student feature $\mathbf{F}^s$ and a teacher feature $\mathbf{F}^t$, hint-based distillation is achieved as follows:

$$\mathcal{L}_{\text{hint}} = \mathcal{D}_{\text{hint}}(\mathsf{T}_{\mathsf{s}}(\mathbf{F}^s), \mathsf{T}_{\mathsf{t}}(\mathbf{F}^t)), \tag{2}$$

where $\mathsf{T}_{\mathsf{s}}$ and $\mathsf{T}_{\mathsf{t}}$ are transformation modules for aligning two features. $\mathcal{D}_{\text{hint}}$ is a measurement of feature divergence, *e.g.*, $l_1$ or $l_2$-norm. In common practice, the hint-based loss is typically used in conjunction with the classification loss, and we adhere to this setting in our experiments.

### 2.2 Vanilla KD can not achieve satisfactory results on small-scale dataset

Recently, many studies have used simplified evaluation protocols that involve small models or datasets. However, there is a growing concern [35] about the limitations of evaluating KD methods solely in such small-scale settings, as many real-world applications involve larger models and datasets. In this section, we extensively investigate the impact of using small-capacity models and small-scale datasets on different KD methods. To provide a comprehensive analysis, we specifically compare vanilla KD with two state-of-the-art logits-based KD approaches, *i.e.*, DKD [19] and DIST [18]. Different from [35] which compresses large models to Res50, our primary focus is to ascertain whether the unsatisfactory performance of vanilla KD can be attributed to small student models or small-scale datasets.

**Impact of limited model capacity.** We conduct experiments using two commonly employed teacher-student pairs: Res34-Res18 and Res50-MobileNetV2. The corresponding results are shown in Table 1a. Previously, these models were trained using the SGD optimizer for 90 epochs ("Original" in table). However, to fully explore the power of KD approaches, we leverage a stronger training strategy by employing the AdamW optimizer with 300 training epochs ("Improve" in table), details can be found in appendix. The original results reported in DKD and DIST demonstrate a clear performance advantage over vanilla KD. However, when adopting a stronger training strategy, all three approaches exhibit improved results. Notably, the performance gap between vanilla KD and the baselines surprisingly diminishes, indicating that *the limited power of vanilla KD can be attributed to insufficient training*, rather than to models with small capacity.

**Impact of small dataset scale.** To investigate the impact of small dataset scale on the performance of vanilla KD, we conduct experiments using the widely used CIFAR-100 dataset [25]. Similarly, we design a stronger training strategy by extending the training epochs from 240 to 2400 and introduce additional data augmentation schemes. As shown in Table 1b, despite the improved strategy leads to better results for all three approaches again, there still remains a performance gap between vanilla KD and other baselines. Notably, the accuracy gap even increases from 1.31% to 2.17% for the Res56-Res20 teacher-student pair. These observations clearly indicate that the underestimation of vanilla KD is not solely attributed to insufficient training, but also to *the small-scale dataset*.

**Discussion.** CIFAR-100 is widely adopted as a standard benchmark for evaluating most distillation approaches. However, our experimental results show that the true potential of vanilla KD has been underestimated on such small-scale datasets. We refer to this phenomenon as the *small data pitfall*, where performance improvements observed on small-scale datasets do not necessarily translate to

Table 1: Impact of dataset scale on KD performance. When adopting a stronger training strategy on large-scale dataset, vanilla KD [15] can achieve on par result to the state-of-the-art approaches, *i.e.*, DKD [19] and DIST [18]. However, this phenomenon is not observed on small-scale dataset, revealing the *small data pitfall* that underestimates the power of vanilla KD. The symbol $\Delta$ represents the accuracy gap between vanilla KD and the best result achieved by other approaches (marked with gray). 'previous recipe' refers to results reported in original papers, while 'stronger recipe' refers to results obtained through our enhanced training strategy.

| Teacher - Student | Method | previous recipe | stronger recipe |
|---|---|---|---|
| Res34 - Res18 (73.31) (69.75) | DKD | 71.70 | 73.07 |
| | DIST | 72.07 | 73.52 |
| | vanilla KD | 70.66 | 73.46 |
| | $\Delta$ | - 1.41 | **- 0.06** |
| Res50 - MBNetV2 (76.16) (68.87) | DKD | 72.05 | 73.71 |
| | DIST | 73.24 | 74.26 |
| | vanilla KD | 68.58 | 74.23 |
| | $\Delta$ | - 4.66 | **- 0.03** |

(a) Vanilla KD achieves comparable results with the state-of-the-art approach on **ImageNet-1K** [30].

| Teacher - Student | Method | previous recipe | stronger recipe |
|---|---|---|---|
| Res56 - Res20 (72.34) (69.09) | DKD | 71.97 | 73.10 |
| | DIST | 71.78 | 74.51 |
| | vanilla KD | 70.66 | 72.34 |
| | $\Delta$ | - 1.31 | - 2.17 |
| Res32×4 - Res8×4 (79.42) (72.50) | DKD | 76.32 | 78.15 |
| | DIST | 75.79 | 77.69 |
| | vanilla KD | 73.33 | 75.90 |
| | $\Delta$ | - 2.99 | - 2.25 |

(b) The performance gap persists on **CIFAR-100** [25], even with stronger training strategy.

more complex real-world datasets. As shown in Figure 1, the performance gap between vanilla Kd and other approaches gradually diminishes with the increasing scale of benchmarks. Considering the elegantly simple of vanilla KD, more extensive experiments are required to explore its full potential.

## 3 Evaluate the power of vanilla KD on large-scale dataset

### 3.1 Experimental setup

In order to fully understand and tap into the potential of vanilla KD, we conduct experiments based on large-scale datasets and strong training strategies. The experimental setup is as following.

**Datasets.** We mainly evaluate KD baselines on the larger and more complex ImageNet-1K dataset [30], which comprises 1.2 million training samples and 50,000 validation samples, spanning across 1000 different categories. Compared to CIFAR-100 dataset [25], ImageNet-1K provides a closer approximation to real-world data distribution, allowing for a more comprehensive evaluation.

**Models.** In our experiments, we mainly utilize Res50 [2] as the student model and BEiTv2-L [43] as the teacher model, which is pretrained on ImageNet-1K and then finetuned on ImageNet-21K and ImageNet-1K successively. This choice is motivated by teacher's exceptional performance, as it currently stands as the best available open-sourced model in terms of top-1 accuracy when using an input resolution of 224x224. In addition, we also include ViT [3, 32] and ConvNeXtV2 [34] as student models, and ResNet [2] and ConvNeXt [5] as teacher models. Specifically, when training ViT student, we do not use an additional distillation token [32].

**Training strategy.** We leverage two training strategies based on recent work [36] that trains high-performing models from scratch using sophisticated training schemes, such as more complicated data augmentation and more advanced optimization methods [32]. Strategy "A1" is slightly stronger than "A2" as summarized in Table 2, and we use it with longer training schedule configurations.

**Baseline distillation methods.** To make a comprehensive comparison, we adopt several recently proposed KD methods as the baselines, such as logits-based vanilla KD [15], DKD [19], and DIST [18], hint-based CC [22], RKD [20], CRD [21], and ReviewKD [17].

### 3.2 Logits-based methods consistently outperform hint-based methods

In this section, we conduct a comparative analysis between logits-based and hint-based distillation approaches. We utilize the widely adopted Res50 architecture [2] as the student model, with Res152 and BEiTv2-L serving as the teachers. The student models are distilled with two different epoch configurations: 300 and 600 epochs. The evaluation results, along with the total GPU training time for each student, are presented in Table 3. To gain a deeper understanding of the generalization of student

Table 2: Training strategies used for distillation. "A1" and "A2" represent two strategies following [36], The upper table provides the shared settings, while the bottom presents their specific settings.

| Setting | T. Res. | S. Res. | BS | Optimizer | LR | LR decay | Warmup ep. | AMP | EMA | Label Loss |
|---|---|---|---|---|---|---|---|---|---|---|
| Common | 224 | 224 | 2048 | LAMB | 5e-3 | cosine | 5 | ✓ | × | BCE |

| Setting | WD | Smoothing | Drop path | Repeated Aug. | H. Flip | PRC | Rand Aug. | Mixup | Cutmix |
|---|---|---|---|---|---|---|---|---|---|
| A2 | 0.03 | × | 0.05 | 3 | ✓ | ✓ | 7/0.5 | 0.1 | 1.0 |
| A1 | 0.01 | 0.1 | 0.05 | 3 | ✓ | ✓ | 7/0.5 | 0.2 | 1.0 |

Table 3: Comparison between hint-based and logits-based distillation methods. The student is Res50, while the teachers are Res152 and BEiTv2-L with top-1 accuracy of 82.83% and 88.39%, respectively. "✓" indicates that the corresponding method is hint-based. GPU hours are evaluated on a single machine with 8 V100 GPUs.

| Scheme | Method | Hint | Epoch | GPU hours | Res152 teacher IN-1K | IN-Real | IN-V2 | BEiTv2-L teacher IN-1K | IN-Real | IN-V2 |
|---|---|---|---|---|---|---|---|---|---|---|
| A2 (79.80) | CC [22] | ✓ | 300 | 265 | 79.55 | 85.41 | 67.52 | 79.50 | 85.39 | 67.82 |
| | RKD [20] | ✓ | 300 | 282 | 79.53 | 85.18 | 67.42 | 79.23 | 85.14 | 67.60 |
| | CRD [21] | ✓ | 300 | 307 | 79.33 | 85.25 | 67.57 | 79.48 | 85.28 | 68.09 |
| | ReviewKD [17] | ✓ | 300 | 439 | 80.06 | 85.73 | 68.85 | 79.11 | 85.41 | 67.36 |
| | DKD [19] | × | 300 | 265 | 80.49 | 85.36 | 68.65 | 80.77 | **86.55** | 68.94 |
| | DIST [18] | × | 300 | 265 | **80.61** | **86.26** | **69.22** | 80.70 | 86.06 | 69.35 |
| | vanilla KD [15] | × | 300 | 264 | 80.55 | 86.23 | 69.03 | **80.89** | 86.32 | **69.65** |
| A1 (80.38) | CC [22] | ✓ | 600 | 529 | 80.33 | 85.72 | 68.82 | - | - | - |
| | RKD [20] | ✓ | 600 | 564 | 80.38 | 85.86 | 68.38 | - | - | - |
| | CRD [21] | ✓ | 600 | 513 | 80.18 | 85.75 | 68.32 | - | - | - |
| | ReviewKD [17] | ✓ | 600 | 877 | 80.76 | 86.41 | 69.31 | - | - | - |
| | DKD [19] | × | 600 | 529 | 81.31 | 86.76 | 69.63 | **81.83** | **87.19** | **70.09** |
| | DIST [18] | × | 600 | 529 | 81.23 | 86.72 | **70.09** | 81.72 | 86.94 | 70.04 |
| | vanilla KD [15] | × | 600 | 529 | **81.33** | **86.77** | 69.59 | 81.68 | 86.92 | 69.80 |
| | DKD [19] | × | 1200 | 1059 | **81.66** | **87.01** | 70.42 | **82.28** | **87.41** | **71.10** |
| | DIST [18] | × | 1200 | 1058 | 81.65 | 86.92 | 70.43 | 81.82 | 87.10 | 70.35 |
| | vanilla KD [15] | × | 1200 | 1057 | 81.61 | 86.86 | **70.47** | 82.27 | 87.27 | 70.88 |

models beyond the ImageNet-1K dataset, we extend the evaluation to include ImageNet-Real [44] and ImageNet-V2 matched frequency [45], which provide separate test sets.

After 300 epochs of training using strategy A2, all hint-based distillation methods exhibit inferior results compared to the logits-based baselines. Despite an extended training schedule with the stronger strategy A1, a notable performance gap remains between the two distillation categories. Moreover, it is worth noticing that hint-based approaches necessitate significantly more training time, highlighting their limitations in terms of both effectiveness and efficiency.

**Discussion.** Our experiments demonstrate that logits-based methods consistently outperform hint-based methods in terms of generalizability. We speculate that this discrepancy can be attributed to the different capabilities of the teacher and student models when dealing with complex distributions. The hint-based approach mimicking the intermediate features of the teacher model becomes less suitable, hindering it from achieving satisfactory results. Furthermore, hint-based methods may encounter difficulties when using heterogeneous teacher and student architectures due to distinct learned representations, which impede the feature alignment process. To analyze this, we perform a center kernel analysis (CKA) [46] to compare the features extracted by Res50 with those of Res152 and BEiTv2-L. As depicted in Figure 2, there is a noticeable dissimilarity between the intermediate

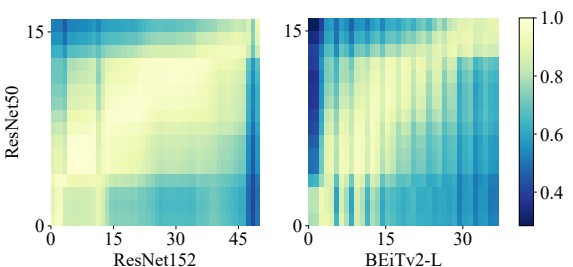

Figure 2: Feature similarity heatmap measured by CKA. *Left*: similarity between homogeneous architectures; *Right*: similarity between heterogeneous architectures. The coordinate axes represent layer indexes.

Table 4: Ablation of different losses used to learn from hard label and soft label in vanilla KD. The "CE", "BCE", and "KL" refers to cross-entropy loss, binary cross-entropy loss, and KL divergence measurement, respectively. BKL is a binary counterpart of KL. Its definition is provided in Equation 3.

| Epoch | Hard Label | Soft Label | Acc. (%) |
|---|---|---|---|
| 300 | CE | KL | 80.49 |
| 300 | BCE | KL | **80.89** |
| 300 | - | KL | 80.87 |
| 300 | CE | BKL | 79.42 |
| 300 | BCE | BKL | 79.36 |
| 300 | - | BKL | 80.82 |
| 600 | BCE | KL | **81.68** |
| 600 | - | KL | 81.65 |

Table 5: Ablation about different configurations of learning rate and weight decay on ImageNet-1K. LR: learning rate; WD: weight decay.

| LR | WD | DKD [19] | DIST [18] | KD [15] |
|---|---|---|---|---|
| 2e-3 | 0.02 | 79.50 | 79.69 | 79.77 |
| 2e-3 | 0.03 | 79.52 | 79.58 | 79.78 |
| 3e-3 | 0.02 | 80.18 | 80.32 | 80.46 |
| 3e-3 | 0.03 | 80.24 | 80.41 | 80.37 |
| 5e-3 | 0.01 | 80.73 | 80.62 | 80.82 |
| 5e-3 | 0.02 | 80.76 | 80.71 | 80.82 |
| 5e-3 | 0.03 | **80.81** | **80.79** | **80.89** |
| 5e-3 | 0.04 | 80.72 | 80.39 | 80.77 |
| 6e-3 | 0.02 | 80.58 | 80.42 | 80.89 |
| 7e-3 | 0.02 | 80.60 | 80.20 | 80.84 |
| 8e-3 | 0.02 | 80.51 | 80.32 | 80.61 |
| 8e-3 | 0.03 | 80.52 | 80.46 | 80.66 |
| 8e-3 | 0.04 | 80.55 | 80.18 | 80.61 |

features of BEiTv2-L and those of the Res50 student, while the Res152 features exhibit a closer resemblance to Res50. In addition to the suboptimal performance, the CKA figure further highlights the incompatibility of hint-based approaches with heterogeneous scenarios, thereby limiting their practical applicability in real-world settings. Consequently, we focus solely on logits-based distillation approaches in subsequent experiments.

### 3.3 Vanilla KD preserves strength with increasing teacher capacity

Table 3 also shows the comparison between vanilla KD and other two logits-based baselines using a stronger heterogeneous BEiTv2-L teacher [43]. This teacher has achieved the state-of-the-art top-1 accuracy on ImageNet-1K validation among open-sourced models.

In all the evaluated settings, the three logits-based approaches exhibit similar performance. After being distilled for 300 epochs with Res152 as the teacher, vanilla KD achieves a top-1 accuracy of 80.55% on ImageNet-1K validation set, which is only 0.06% lower than the best student performance achieved by DIST. With an extended training duration of 600 epochs, the student trained using vanilla KD even achieves the top performance at 81.33%. When taught by BEiT-L, vanilla KD obtains 80.89% top-1 accuracy, surpassing both DKD and DIST. Even in the presence of distribution shift, vanilla KD maintains comparable performance to other approaches on ImageNet-Real and ImageNet-V2 matched frequency benchmarks. Additionally, further extending the training schedule to 1200 epochs improves vanilla KD and results in a performance on par with DKD and DIST, thus highlighting its effectiveness in delivering impressive results.

These results clearly indicate that vanilla KD has been underestimated in previous studies because they are designed and evaluated on small-scale datasets. When trained with both powerful homogeneous and heterogeneous teachers, vanilla KD achieves competitive performance that is comparable to state-of-the-art methods while still maintaining its simplicity.

### 3.4 Robustness of vanilla KD across different training components

In this section, we perform ablation studies to assess the robustness of vanilla KD under different training settings. Specifically, we compare various loss functions used in vanilla KD and explore different hyper-parameter configurations for the three logits-based distillation methods mentioned earlier.

**Ablation on Loss functions.** Since our strategies [36] focus on training the student using one-hot labels and the binary cross-entropy (BCE) loss function, we also evaluate a binary version of the KL divergence as an alternative for learning from soft labels:

$$\mathcal{L}_{\text{BKL}} = \sum_{i \sim \mathcal{Y}} [-\boldsymbol{p}_i^t \log(\boldsymbol{p}_i^s / \boldsymbol{p}_i^t) - (1 - \boldsymbol{p}^t) \log((1 - \boldsymbol{p}^s)/(1 - \boldsymbol{p}^t))], \qquad (3)$$

where $\mathcal{Y}$ denotes the label space.

Table 6: Distillation results of logits-based approaches with various teacher and student combinations. Number in the parenthesis reports the result of training corresponding model from scratch.

| LR | WD | ConvNeXt-XL - Res50 (86.97) (79.86) | | | LR | WD | BEiTv2-L - DeiT-S (88.39) (79.90) | | | ConvNeXt-XL - DeiT-S (86.97) (79.90) | | |
|---|---|---|---|---|---|---|---|---|---|---|---|---|
| | | DKD | DIST | KD | | | DKD | DIST | KD | DKD | DIST | KD |
| 3e-3 | 0.02 | 80.50 | 80.50 | 80.71 | 3e-4 | 0.05 | 80.11 | 79.44 | 80.45 | 80.18 | 79.73 | 80.45 |
| 5e-3 | 0.02 | **81.05** | **80.89** | 80.94 | 5e-4 | 0.03 | 79.82 | 79.52 | 80.28 | 80.17 | 79.61 | 80.19 |
| 5e-3 | 0.03 | 81.02 | 80.85 | 80.98 | 5e-4 | 0.05 | 80.55 | 79.58 | 80.87 | 80.56 | 80.19 | 80.89 |
| 5e-3 | 0.04 | 80.90 | 80.84 | **81.10** | 5e-4 | 0.07 | **80.80** | **80.00** | **80.99** | 80.86 | 80.26 | 80.87 |
| 7e-3 | 0.02 | 80.81 | 80.86 | 81.07 | 7e-4 | 0.05 | 80.53 | 79.94 | 80.97 | 80.80 | 80.25 | **80.99** |

We compare various combinations of loss functions for vanilla KD and present the results in Table 4. Surprisingly, the binary cross-entropy (BCE) loss outperforms the multi-class cross-entropy loss in terms of accuracy. However, we find that the binary version of KL divergence, referred to as BKL loss, yields suboptimal performance. Moreover, the results indicate that training with or without supervision from hard labels has minimal impact on performance, even with a longer training duration. Consequently, we maintain hard label supervision [15, 19] throughout our experiments.

**Ablation on training hyper-parameters.** To ensure a thorough and fair comparison, we investigate various configurations of learning rate and weight decay for DKD, DIST, and vanilla KD using strategy "A2". The results, presented in Table 5, reveal an intriguing finding: vanilla KD outperforms the other methods across all configurations. In addition, it is noteworthy that vanilla KD consistently performs close to its best across different learning rates and weight decay configurations, demonstrating its robustness and ability to maintain competitive results under various settings.

## 3.5 Distillation results with different teacher-student pairs

In addition to the BEiTv2-L teacher and the Res50 student used in previous experiments, we introduce additional combinations of teacher-student pairs: ConvNeXt-XL-Res50, BEiTv2-L-DeiT-S, and ConvNeXt-XL-DeiT-S. We conduct experiments to compare the performance of vanilla KD with that of DKD and DIST on these combinations. Each student model is trained for 300 epochs using strategy "A2". We evaluate multiple hyper-parameter configurations, including learning rate and weight decay, and select the configuration with the highest performance for each method to ensure a fair comparison. Corresponding results are presented in Table 6. Consistent with the findings from previous experiments, vanilla KD achieves on par or slightly better performances compared to DKD and DIST on all combinations. These results demonstrate the potential of vanilla KD to achieve state-of-the-art performances across a diverse range of teacher-student combinations.

**Marginal gains when teacher is sufficiently large.** Although vanilla KD can achieve state-of-the-art results across various architectures in our experiments, it is not without limitations. For example, we observe that the Res50 student, trained with a BEiTv2-B teacher, achieves similar or even better result than that trained with a larger BEiTv2-L. This observation is consistent with the analysis presented in [47]. To gain deeper insights into the impact of teacher model when working with large-scale datasets, we conducted additional experiments in this section. We evaluate the impact of the BEiTv2-B teacher and the BEiTv2-L teacher with all the logits-based baselines, using three different epoch configurations. As shown in Table 9, the results demonstrate a trend of performance degradation as the capacity of the teacher model increases. This indicates that vanilla KD is not able to obtain benefit from a larger teacher model, even trained on a large-scale dataset.

**Sustained benefits when enlarging the teacher's training set.** we also evaluate the influence of the training set used for teacher models on student performance. We compare two BEiTv2-B teacher models: one pre-trained on ImageNet-1K and the other pre-trained on ImageNet-21K. Subsequently, we train two distinct students using strategy "A1" for 1200 epochs. The evaluation results are presented in Table 7. The results clearly demonstrate that when the teacher is trained on a larger-scale dataset, it positively affects the performance of the student. We hypothesize that

Table 7: Teacher trained with larger dataset can produce a better student. The teacher is BEiTv2-B.

| Student | IN-1K T. (85.60) | IN-21K T. (86.53) |
|---|---|---|
| DeiT-T | 76.12 | 77.19 |
| Res50 | 81.97 | 82.35 |

Table 9: Impact of different teacher model sizes. The student model is Res50.

| Method | Teacher | 300 Ep. | 600 Ep. | 1200 Ep. |
|--------|---------|---------|---------|----------|
| DKD | BEiTv2-B | 80.76 | 81.68 | **82.31** |
|     | BEiTv2-L | **80.77** | **81.83** | 82.28 |
| DIST | BEiTv2-B | **80.94** | **81.88** | **82.33** |
|      | BEiTv2-L | 80.70 | 81.72 | 81.82 |
| KD | BEiTv2-B | **80.96** | 81.64 | **82.35** |
|    | BEiTv2-L | 80.89 | **81.68** | 82.27 |

Table 10: Vanilla KD *vs.* FCMIM on ConveNeXtV2-T. [†]: the GPU hours of fine-tuning stage is 568.

| Method | EP. | Hours | Acc. (%) |
|--------|-----|-------|----------|
| FCMIM (IN-1K) | 800 | 1334 | - |
| + Fine-tune on IN-1K | 300 | 1902[†] | 82.94 |
| FCMIM (IN-1K) | 800 | 1334 | - |
| + Fine-tune on IN-21K | 90 | 3223 | - |
| + Fine-tune on IN-1K | 90 | 3393 | 83.89 |
| vanilla KD | 300 | 653 | 84.42 |
| vanilla KD | 1200 | 2612 | **85.03** |

a teacher pre-trained on a larger dataset has a better understanding of the data distribution, which subsequently facilitates student learning and leads to improved performance.

## 3.6 Exploring the power of vanilla KD with longer training schedule

In Section 3.3, we notice a consistent improvement in the performance of vanilla KD as the training schedule is extended. To explore its ultimate performance, we further extended the training schedule to 4800 epochs. In this experiment, we utilized a BEiT-B teacher model, which achieved 86.47% top-1 accuracy on the ImageNet-1K validation set. As for students, we employed Res50 [2], ViT-T, ViT-S [3, 32], and ConveNextV2-T [34], and trained

Table 8: Results of extended training schedule. The teacher model is BEiTv2-B.

| Model \ Epoch | 300 | 600 | 1200 | 4800 |
|---------------|-----|-----|------|------|
| Res50 | 80.96 | 81.64 | 82.35 | **83.08** |
| ViT-S | 81.38 | 82.71 | 83.79 | **84.33** |
| ViT-T | - | - | 77.19 | **78.11** |
| ConvNextV2-T | 84.42 | 84.74 | **85.03** | - |

them using strategy "A1". The corresponding results are shown in Table 8. All three students achieve improved performance with longer training epochs. This trend is also consistent with the pattern observed in [35]. When trained with 4800 epochs, the students achieve new state-of-the-art performance, surpassing previous literature [35, 32]. Specifically, the Res50, ViT-S, ViT-T, and ConvNeXtV2-T achieve 83.08%, 84.33%, 78.11% and 85.03%[2] top-1 accuracy, respectively.

## 3.7 Comparison with masked image modeling

Masked image modeling (MIM) framework [48, 49, 50] has shown promising results in training models with excellent performance. However, it is worth noting that the pre-training and fine-tuning process of MIM can be time-consuming. In this section, we conduct a comparison between MIM and vanilla KD, taking into account both accuracy and time consumption. Specifically, we compare vanilla KD with the FCMIM framework proposed in ConveNeXtv2 [34]. We utilize a ConvNeXtv2-T student model and a BEiTv2-B teacher model. We train two student models using vanilla KD: one for 300 epochs and another for 1200 epochs.

The results are presented in Table 10, highlighting the efficiency of vanilla KD in training exceptional models. Specifically, even with a training duration of only 300 epochs, vanilla KD achieves an accuracy of 84.42%, surpassing MIM by 0.53%, while consuming only one-fifth of the training time. Moreover, when extending the training schedule to 1200 epochs, vanilla KD achieves a performance of 85.03%, surpassing the best-performing MIM-trained model by 1.14%. These results further demonstrate the effectiveness and time efficiency of vanilla KD compared to the MIM framework.

## 3.8 Transferring to downstream task

To assess the transfer learning performance of the student distilled on ImageNet, we conduct experiments on object detection and instance segmentation tasks using COCO benchmark [26]. We adopt Res50 and ConvNeXtV2-T as backbones, and initialize them with the distilled checkpoint. The detectors in our experiments are commonly used Mask RCNN [40] and Cascade Mask RCNN [41].

---

[2]Here we find that the result of KD has already reached a satisfactory level, and the training cost is lower compared to the MIM approach. Hence, we did not pursue further scaling to 4800 epochs.

Table 11: Object detection and instance segmentation results on COCO. We compare checkpoints with different pre-trained accuracies on ImageNet when used as backbones in downstream tasks.

| (a) Mask RCNN (Res50 [2]). | | | | | (b) Mask RCNN (ConvNeXtV2-T [34]). | | | | | (c) Cascade Mask RCNN (ConvNeXtV2-T [34]). | | | | |
|---|---|---|---|---|---|---|---|---|---|---|---|---|---|---|
| ckpt. | sche. | IN-1K | AP$^b$ | AP$^m$ | ckpt. | sche. | IN-1K | AP$^b$ | AP$^m$ | ckpt. | sche. | IN-1K | AP$^b$ | AP$^m$ |
| prev. | 1× | 77.1 | 38.2 | 34.7 | prev. | 1× | 83.0 | 45.4 | 41.5 | prev. | 1× | 83.0 | 49.8 | 43.5 |
| prev. | 1× | 80.4 | 38.7 | 35.1 | prev. | 1× | 83.9 | 45.6 | 41.6 | prev. | 1× | 83.9 | 50.4 | 44.0 |
| ours | 1× | 83.1 | 41.8 | 37.7 | ours | 1× | 85.0 | 45.7 | 42.0 | ours | 1× | 85.0 | 50.6 | 44.3 |
| prev. | 2× | 77.1 | 39.2 | 35.4 | prev. | 3× | 83.0 | 47.4 | 42.7 | prev. | 3× | 83.0 | 51.1 | 44.5 |
| prev. | 2× | 80.4 | 40.0 | 36.1 | prev. | 3× | 83.9 | 47.7 | 42.9 | prev. | 3× | 83.9 | 51.7 | 44.9 |
| ours | 2× | 83.1 | 42.1 | 38.0 | ours | 3× | 85.0 | 47.9 | 43.3 | ours | 3× | 85.0 | 52.1 | 45.4 |

Table 11 reports the corresponding results. When utilizing our distilled Res50 model as the backbone, Mask RCNN outperforms the best counterpart using a backbone trained from scratch by a significant margin (+3.1% box AP using "1×" schedule and +2.1% box AP using "2×" schedule). Similarly, when the backbone architecture is ConvNeXtV2-T, using the model trained by vanilla KD consistently leads to improved performance. These results demonstrate the efficient transferability of the performance improvements achieved by vanilla KD to downstream tasks.

## 4   Related work

**Knowledge distillation (KD).** Previous KD methods can be broadly categorized into two classes: logits-based methods and hint-based methods. Hinton *et al.* [15] first proposes to train the student to learn from logits of teacher. This method is referred as vanilla KD. Recently, researchers have introduced improvements to the vanilla KD method by enhancing the dark knowledge [19, 51], relaxing the constraints [18], using sample factors to replace the unified temperature [52]. Hint-based methods train the student to imitate learned representations of the teacher. These methods leverage various forms of knowledge, such as model activation [16, 53, 54], attention map [55], flow of solution procedure [56], or feature correlation among samples [22, 20]. To facilitate the alignment of features between the teacher and the student, techniques including design new projector [57, 58] and more complex alignment approaches [17, 59, 60] have been proposed. Besides directly mimicking the hint knowledge, contrastive learning [21] and masked modeling [61] have also been explored. Recently, the success of vision transformers [3] has prompted the development of specific distillation approaches tailored to this architecture, including introducing additional distillation tokens [32, 62] or perform distillation at the fine-grained patch level [63]. Additionally, vision transformer architectures have also been leveraged to facilitate KD between CNN models [64, 65]. Furthermore, some works attempt to address specific challenges in KD, such as distilling from ensembles of teachers [66, 67] and investigating the impact of different teachers on student performance [68, 47, 69].

**Large-scale training.** Recent developments in large-scale deep learning [31, 48] suggest that by scaling up data [3, 70], model size [6, 7, 8] and training iteration [35] to improve model performance, achieve state-of-the-art results [71] across diverse tasks, and enable better adaptability to real-world scenarios. For example, in computer vision, numerous large models have achieved outstanding results on tasks such as classification, detection, and segmentation. These models are usually trained on extensive labeled image datasets, including ImageNet-21K [30] and JFT [15, 72]. Similarly, in natural language processing, training models on massive text corpora [73, 74] has led to breakthroughs in language understanding, machine translation, and text generation tasks. Inspired by this scaling trend, it becomes crucial to evaluate KD approaches in more practical scenarios. After all, the goal of KD is to enhance the performance of student networks.

The closest work to ours is that of Beyer *et al.* [35]. They identify several design choices to make knowledge distillation work well in model compression. Taking it a step further, we point out the small data pitfall in current distillation literature: small-scale datasets limit the power of vanilla KD. We thoroughly study the crucial factors in deciding distillation performance. In addition, we have distilled several new state-of-the-art backbone architectures, further expanding the repertoire of high-performing models in vision arsenal.

# 5   Conclusion and discussion

Motivated by the growing emphasis on *scaling up* for remarkable performance gains in deep learning, this paper revisits several knowledge distillation (KD) approaches, spanning from small-scale to large-scale dataset settings. Our analysis has disclosed the small data pitfall present in previous KD methods, wherein the advantages offered by most KD variants over vanilla KD diminish when applied to large-scale datasets. Equipped with enhanced data augmentation techniques and larger datasets, vanilla KD emerges as a viable contender, capable of achieving comparable performance to the most advanced KD variants. Remarkably, our vanilla KD-trained ResNet-50, ViT-S, and ConvNeXtV2-T models attain new state-of-the-art performances without any additional modifications.

While the application of knowledge distillation on large-scale datasets yields significant performance improvements, it is important to acknowledge the associated longer training times, which inevitably contribute to increased carbon emissions. These heightened computational resource requirements may impede practitioners from thoroughly exploring and identifying optimal design choices. Future endeavors should focus on limitations posed by longer training times and carbon emissions, and exploring new avenues to further enhance the efficacy and efficiency of distillation methods.

## Acknowledgement

This work is supported by National Key Research and Development Program of China under No. SQ2021YFC3300128 National Natural Science Foundation of China under Grant 61971457. And Chang Xu was supported in part by the Australian Research Council under Projects DP210101859 and FT230100549.

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
