# Revisit the Power of Vanilla Knowledge Distillation: from Small Scale to Large Scale Supplementary Material

**Zhiwei Hao**[1,2*]**, Jianyuan Guo**[3*]**, Kai Han**[2]**, Han Hu**[1†]**, Chang Xu**[3]**, Yunhe Wang**[2†]

[1]School of information and Electronics, Beijing Institute of Technology.
[2]Huawei Noah's Ark Lab.
[3]School of Computer Science, Faculty of Engineering, The University of Sydney.

{haozhw, hhu}@bit.edu.cn, jguo5172@uni.sydney.edu.au,
{kai.han, yunhe.wang}@huawei.com, c.xu@sydney.edu.au

## A  Impact of small-scale dataset

### A.1  Details of "stronger recipe"

In Table 1 of our main paper, we evaluate the impact of limited model capacity [1] and small-scale dataset by comparing the results of using "previous training recipe" and our "stronger recipe". We summarize the details of "stronger recipe" and present them in Table 13.

Table 13: Stronger training strategy used for distillation. "B" and "C" represent strategies for training students on ImageNet-1K and CIFAR100, respectively.

| Setting | T. Res. | S. Res. | BS | Optimizer | LR | LR decay | Warmup epochs | AMP | Label Loss |
|---------|---------|---------|------|-----------|------|----------|---------------|-----|------------|
| B | 224 | 224 | 1024 | AdamW | 1e-3 | cosine | 20 | ✓ | CE |
| C | 32 | 32 | 512 | SGD | 5e-2 | cosine | × | × | CE |

| Setting | WD | H. Flip | Random erasing | Auto Augment | Rand Augment | Mixup | Cutmix |
|---------|------|---------|----------------|--------------|--------------|-------|--------|
| B | 5e-2 | ✓ | 0.25 | × | 7/0.5 | 0.1 | 1.0 |
| C | 5e-4 | ✓ | × | ✓ | × | 0.1 | × |

### A.2  Numerical results

In Figure 1 of our main paper, we present a comparison of performance gaps among vanilla KD and two logits-based baselines, *i.e.*, DKD [2] and DIST [3], on two datasets of varying scales, to demonstrate the underestimation of vanilla KD on small-scale datasets. The benchmarks used for this comparison include CIFAR-100, ImageNet-1K and two subsets of ImageNet-1K. These subsets are obtained by stratified sampling from the training set of ImageNet-1K with fractions of 30% (0.38M training images) and 60% (0.77M training images), respectively. Table 14 reports numerical results of the comparison. On CIFAR-100 dataset, students are trained for 2400 epochs with strategy "C", while on ImageNet-1K dataset and its subsets, *i.e.*, 30%/60%/full, students are trained for 4000/2000/1200 epochs (same total iteration) with strategy "A1".

---

[*]Equal contribution.  [†]Corresponding author.

37th Conference on Neural Information Processing Systems (NeurIPS 2023).

Table 14: Comparison of different dataset scales. Results in parentheses report the performance improvement over vanilla KD.

| Dataset | CIFAR-100 | IN-1K (30%) | IN-1K (60%) | IN-1K (full) |
|---|---|---|---|---|
| Teacher | Res56 | BEiTv2-B | BEiTv2-B | BEiTv2-L |
| Student | Res20 | Res50 | Res50 | Res50 |
| vanilla KD | 72.34 | 79.58 | 81.33 | 82.27 |
| DKD | 73.10 (+0.76) | 79.86 (+0.28) | 81.47 (+0.14) | 82.28 (+0.01) |
| DIST | 74.51 (+2.17) | 79.75 (+0.17) | 81.34 (+0.01) | 81.82 (- 0.46) |

Table 15: Results on ImageNet-22K. The teacher is BEiTv2-B and the student is Res50.

| Method | IN-21K iter. | IN-1K iter. | Total iter. | Acc. (%) |
|---|---|---|---|---|
| DKD | 0 | 375.0K | 375.0K | 81.68 |
| DIST | 0 | 375.0K | 375.0K | 81.88 |
| vanilla KD | 0 | 375.0K | 375.0K | 81.64 |
| vanilla KD | 0 | 750.0K | 750.0K | 82.35 |
| vanilla KD | 0 | 3000K | 3000K | 83.08 |
| DKD | 187.5K | 187.5K | 375.0K | 81.21 |
| DIST | 187.5K | 187.5K | 375.0K | 81.20 |
| vanilla KD | 187.5K | 187.5K | 375.0K | 81.29 |
| vanilla KD | 187.5K | 562.5K | 750.0K | 81.98 |
| vanilla KD | 750K | 3000K | 3750K | 82.85 |

# B Scaling up to ImageNet-22K

To further investigate the performance of vanilla KD on datasets with larger scale, we conduct experiments on ImageNet-22K. Our experiment follows a two-stage KD setting. In the first stage, KD is performed exclusively using samples from ImageNet-21K, where the student model learns solely from the soft labels provided by the ImageNet-1K pretrained teacher (the 21K images are classified into 1K labels, avoid re-training a new classify head). In the second stage, distillation is performed on ImageNet-1K similar to our previous experiments.

The results are summarized in the Table 15. Our intuition of utilizing additional samples from ImageNet-21K is to provide the student model with a broader range of data distributions. However, we found that when trained for an equal number of iteration steps, the performance on ImageNet-22K was inferior to that achieved on ImageNet-1K. We hypothesize that this discrepancy arises from the simplistic approach of classifying 21K images into the 1K categories through the pre-trained teacher, which may result in the out-of-distribution problem. Moreover, if we were to follow conventional methods (22K pre-training then 1K fine-tuning) such as re-train a classification head for both teacher and student, it might offer some assistance. However, this approach would significantly increase the computational cost associated with the entire knowledge distillation process, which goes against the original intention of KD to directly leverage existing teacher model. In conclusion, more sophisticated approaches are required to fully harness the potential of the additional out-of-distribution samples.

# C Ablation of hyper-parameters in baseline methods

In our main paper, results of DKD and DIST are obtained using hyper-parameters in their original implementation. The loss function of DKD and DIST are presented as follows:

$$
\begin{aligned}
\mathcal{L}_{\text{DKD}} &= \alpha_{\text{DKD}}\text{TCKD} + \beta_{\text{DKD}}\text{NCKD} \\
\mathcal{L}_{\text{DIST}} &= \mathcal{L}_{\text{cls}} + \beta_{\text{DIST}}\mathcal{L}_{\text{inter}} + \gamma_{\text{DIST}}\mathcal{L}_{\text{intra}}.
\end{aligned}
\tag{1}
$$

By default, hyper-parameters $\alpha_{\text{DKD}}$, $\beta_{\text{DKD}}$, $\beta_{\text{DIST}}$, and $\gamma_{\text{DIST}}$ are set to 1, 2, 1, and 1, respectively. In order to provide a fair evaluation of these logits-based baselines, we conduct experiments to study the impact of different settings of these hyper-parameters. We use the training strategy "A2" and train all models for 300 epochs. The results of the ablation study are presented in Table 16. By modifying the hyper-parameters, the best accuracy of DKD and DIST improves to **80.96%** and

Table 16: Ablation of hyper-parameters in baseline methods on ImageNet-1K. The teacher model is BEiTv2-B and the student model is Res50. The best results are indicated in **bold**. The hyper-parameters adopted in original baselines of our main paper are marked with gray, which is also the default setting in their corresponding paper.

| DKD | | | | DIST | | | |
|---|---|---|---|---|---|---|---|
| $\alpha_{DKD}$ ($\beta_{DKD}=2$) | Acc. (%) | $\beta_{DKD}$ ($\alpha_{DKD}=1$) | Acc. (%) | $\beta_{DIST}$ ($\gamma_{DIST}=1$) | Acc. (%) | $\gamma_{DIST}$ ($\beta_{DIST}=1$) | Acc. (%) |
| 0 | 78.02 | 0.2 | 80.54 | 0.5 | 80.74 | 0 | 80.19 |
| 0.2 | 79.56 | 0.5 | 80.84 | 1 | 80.94 | 0.5 | 80.91 |
| 0.5 | 80.30 | 1 | **80.96** | 2 | **81.01** | 1 | **80.94** |
| 1 | 80.76 | 2 | 80.76 | 3 | 80.74 | 2 | 80.86 |
| 2 | **80.93** | 4 | 80.31 | 4 | 80.64 | - | - |
| 4 | 80.81 | 8 | 79.86 | 5 | 80.64 | - | - |

Table 17: Comparison of different dataset scales on object detection tasks. The used performance metric is mAP. Results in parentheses report the performance improvement over vanilla KD.

| Dataset | Pascal VOC | COCO (30%) | COCO (60%) | COCO (full) |
|---|---|---|---|---|
| from scratch | 42.88 | 27.32 | 30.97 | 33.26 |
| vanilla KD | 46.95 | 30.32 | 32.98 | 34.75 |
| DKD | 48.07 (+1.12) | 31.27 (+0.95) | 33.40 (+0.42) | 35.07 (+0.32) |
| DIST | 47.64 (+0.69) | 31.67 (+1.35) | 33.18 (+0.20) | 34.52 (- 0.23) |

**81.01**% respectively, which is an increase of 0.20% and 0.07% compared to their original settings. However, vanilla KD achieves a comparable performance of **80.96**% top-1 accuracy, without any modifications to its hyper-parameters, as shown in Table 9 of our main paper.

# D  Impact of dataset scale on object detection tasks

To investigate the impact of data on knowledge distillation in the context of object detection tasks [4, 5, 6], we conduct additional experiments on the Pascal VOC [7] and COCO [8] datasets. Specifically, the Pascal VOC dataset consists of 20 object classes. Our training set is the combination of VOC 2007 trainval (5K) and VOC 2012 trainval (11K), with the validation set being VOC 2007 test (4.9K), following previous protocals. The training samples in Pascal VOC (16K) are smaller than those in COCO (118K). Therefore, we will also utilize subsets of COCO (30% and 60%) for conducting experiments. We adopt Faster-RCNN [9] as the detection framework. Backbone architecture of the teacher and the student are ResNet-101 and ResNet-18, respectively.

The experiments on detection tasks cover four datasets: (i) VOC, with 16K training samples and 20 categories; (ii) 30% COCO, with 35K training samples and 80 categories; (iii) 60% COCO, with 71K training samples and 80 categories; and (iv) full COCO, with 118K training samples and 80 categories. From the experimental results shown in Table 17, it can be observed that on datasets with similar difficulty levels, the performance gap between vanilla KD and DKD gradually diminishes as the number of training samples increases. For example, on the 30% COCO dataset, the mAP of DKD is 0.95 higher than that of vanilla KD, while on the full COCO dataset, it is 0.32. This phenomenon is analogous to the "small data pitfall" observed in the classification task, where as the dataset size grows, the performance gap between vanilla KD and other KD methods narrows. Additionally, in both the VOC and 30% COCO settings (the use of relatively small training datasets), despite different task difficulties (categories), the performance gap between vanilla KD and DKD methods remains similar.