# OpenReview forum: "Revisit the Power of Vanilla Knowledge Distillation: from Small Scale to Large Scale"
_NeurIPS.cc/2023/Conference — NeurIPS 2023 poster_

### Official Review · Reviewer_aER7 · 2023-06-14

**Soundness:** 2 fair
**Presentation:** 3 good
**Contribution:** 2 fair
**Rating:** 4
**Confidence:** 5

**Summary:**

This paper explores the power of vanilla distillation on large datasets and strong training recipes. It shows the stronger data augmentation and using larger datasets can decrease the gap between vanilla KD and other meticulously designed methods. The extensive results show vanilla KD's power.

**Strengths:**

1. This paper reveals the enormous potential of vanilla KD. It inspires the community to take a broader look at knowledge distillation instead of small models or datasets.
2. This paper takes an enormous number of experiments to explore vanilla KD and obtain many valuable results and SOTA models.

**Weaknesses:**

1. We usually train the models under an ordinary training recipe actually. And under this setting, vanilla KD performs worse. So it is still not clear which method we should apply for various settings. For example, in LKBT[1], this paper compares KD and another method NKD under various settings including stronger training recipes. But NKD performs much better than KD.

    [1] Are Large Kernels Better Teachers than Transformers for ConvNets. ICML 2023

2. The paper lacks the baseline performance under a stronger recipe. Generally, KD is introduced for a small model and ordinary training recipe. When the training recipe is stronger, the improvements that KD brings are fewer. In this paper, we cannot know how much KD helps the students under a stronger training recipe.
3. The paper applies downstream tasks by replacing the backbones. However, DKD and Dist can be applied to the classification branch directly for detection distillation. They perform much better than KD. How about vanilla KD performs under a stronger training recipe for detection? Does the conclusion keep the same?
4. The hyper-parameter $\alpha$ and temperature for KD are also important. This paper lacks a discussion about this.
5. Some typos, e.g. the performance of Res50 and MBNetV2 are reversed in Tab 1. Some details are missing, for example, the teacher BeiTV2 seems to be training with ImageNet-21K. The paper needs to clarify this.

**Questions:**

see the weakness

**Limitations:**

yes

---

> ### Author Rebuttal · Authors · 2023-08-10
>
> **Response to weakness 1:** Thanks for your valuable comments. We first answer for the method should be applied for different settings. Through our empirical sutdy, we observe a trend wherein a small training set prefers a knowledge distillation (KD) method with stronger regularization or priors, as it can effectively bring in more informative knowledge from the teacher model to enhance the performance of the smaller student model. On the other hand, with a large training set, the need for additional prior diminishes. The sheer abundance of diverse data in a larger dataset already provides sufficient information for the student model to learn from. In such cases, a simpler KD approach may suffice, as the teacher model can effectively capture the complexities present in the data without requiring extensive regularization or priors.
>
> To verify the above analysis, we present the average confidence on all samples and the entropy of predictions of pretrained ResNet50/ResNet152 teacher models on CIFAR-100/ImageNet-1K. We use the two following measurements:
>
> $
> \max({p}(x))=\mathbb{E}_{x\in\mathcal{X}}[\max_i(p_i)],
> $
>
> and
>
> $
> \text{entropy}({p}(x))=\mathbb{E}_{x\in\mathcal{X}}[\textstyle \sum_i p_i\log p_i],
> $
>
> where $p_i$ is predictive probability of sample $x$ belonging to class $i$.
>
> The average confidence score provides an insight into how certain the teacher's predictions are across all samples. As shown in the table below, model trained on CIFAR-100 exhibits higher average confidence and lower entropy value, suggesting that the model is more confident in its predictions, however, resulting in reduced mutual information or knowledge transfer among other classes.
>
> ||Top-1|$\max({p}(x))$|$\text{entropy}({p}(x))$|
> |:-:|:-:|:-:|:-:|
> |CIFAR-100|79.33|89.78|0.3362|
> |ImageNet-1K|75.81|79.63|0.7593|
>
> Next, we address the question of why NKD [1] outperforms KD. Take the SLaK-T - Res50 teacher-student pari as an example, when distilling for 120 epochs on ImageNet, NKD achieves a top-1 accuracy of 78.57%, while KD achieves 77.05%. However, extending the distillation duration to 300 epochs leads to NKD achieving a top-1 accuracy of 80.24%, and in our implementation of this extended training (which was not reported in [1]), KD achieves a top-1 accuracy of 80.11%. Remarkably, the gap between KD and NKD narrows significantly with this increased training epoch, further corroborating our observation that aligns with the concept of small data pitfall.
>
> **Response to weakness 2:** In Table 1 of our main paper, we compared the performance of several baseline models using both ordinary and stronger training recipes. On small-scale datasets, the stronger training recipe improves the performance of vanilla KD, yet a significant performance gap remains between vanilla KD and recent KD baselines. On large-scale datasets, the stronger recipe also leads to improved performance for vanilla KD, resulting in a reduction of the performance gap. Building on the aforementioned observation, we introduce the concept of the "small data pitfall." Furthermore, as a response to ***weakness 1 of oNYG***, we conducted experiments using the lightweight MobileNet V3 Small model, reaffirming the validity of the small data pitfall identified in our main paper. For your convenience, we present the results here:
>
> |Dataset|Teacher|Student|Method|Top-1|
> |:-:|:-:|:-:|:-:|:-:|
> |CIFAR-100|-|MobileNet v3 Small|-|64.76|
> |CIFAR-100|ResNet32x4|MobileNet v3 Small|DKD|**69.10**|
> |CIFAR-100|ResNet32x4|MobileNet v3 Small|DIST|67.72|
> |CIFAR-100|ResNet32x4|MobileNet v3 Small|KD|68.59|
> |ImageNet-1K|-|MobileNetv3Small|-|67.40|
> |ImageNet-1K|BeiTv2-B|MobileNet v3 Small|DKD|67.36|
> |ImageNet-1K|BeiTv2-B|MobileNet v3 Small|DIST|68.02|
> |ImageNet-1K|BeiTv2-B|MobileNet v3 Small|KD|**68.05**|
>
> **Response to weakness 3:** We followed the released DKD code to conduct object detection experiments, directly applying distillation to the classification branch. The results are presented in the following table. We used ResNet101 as the teacher backbone and ResNet18 as the student backbone. Among the results, vanilla KD and DIST achieved similar performance, while DKD outperformed the other two methods. Based on our small data pitfall assumption, we speculate that the performance gap is due to the limited scale of the COCO dataset. With a larger dataset, vanilla KD could potentially achieve more competitive performance. To further validate this, we plan to explore stronger training recipes (mosaic augmentation and large-scale jittering) and larger datasets (such as Objects365) for downstream tasks in our future work.
>
> |Iteration|Method|AP|AP50|AP75|
> |:-:|:-:|:-:|:-:|:-:|
> |180K|From Scratch|33.26|53.61|35.26|
> |180K|DKD|35.07|56.32|37.45|
> |180K|DIST|34.52|55.64|37.33|
> |180K|KD|34.75|55.95|37.31|
> |540K|DKD|37.44|58.75|40.37|
> |540K|DIST|37.01|58.00|40.07|
> |540K|KD|36.99|57.92|39.87|
>
> **Response to weakness 4:** Thanks for the valuable comments. We adopt a trivial setting of hyperparameter $\alpha$ by setting its value to 1, as our primary focus is not on optimizing the configuration of vanilla KD, but rather on showcasing its latent potential. We believe that this straightforward setting is sufficient to support our conclusion. Furthermore, our ablation study, as presented in Table 4, has explored the scenario with $\alpha=0$, which yielded performance similar to that of $\alpha=1$. This suggests that the specific value of $\alpha$ has minimal impact on our main findings.
>
> ***Regarding the temperature parameter, please refer to our "Author Rebuttal" section.***
>
> **Response to weakness 5:** Thanks for your valuable suggestion. The BeiTV2 teacher is trained with ImageNet-21K. More specifically, it is pretrained on ImageNet-1K and then finetuned on ImageNet-21K and ImageNet-1K successively. We will correct the typos and added more details about the BeiTV2 teacher in our final version.

---

> > ### Comment · Reviewer_aER7 · 2023-08-13
> > **Response to Rebuttal.**
> >
> > We thank the author for rebuttals.
> >
> > Some of my concerns are addressed. However, the performance on the downstream tasks, and the baseline performance under a strong training recipe without KD is still not satisfactory.

---

> > > ### Author Response · Authors · 2023-08-13
> > > **Response to Reviewer aER7 part (1/2)**
> > >
> > > Thanks for your nice comments.
> > >
> > > ~~We are conducting more downstream tasks~~.
> > >
> > > We apologize for the oversight in not including the results for the "baseline performance under a strong training recipe without KD" in our previous response. We misunderstood the target of your reference to "baseline". ~~As soon as the ongoing experiments are concluded, we will **promptly update the table below** and incorporate the results into Table 1 of our main paper.~~
> > >
> > > on ImageNet-1K:
> > > |Teacher-Student|Method|previous recipe|effective gain|direct gain|stronger recipe|effective gain|direct gain|
> > > |:-:|:-:|:-:|:-:|:-:|:-:|:-:|:-:|
> > > |Res34-Res18|From Scratch|69.75|-|-|**71.91**|-|-|
> > > |Res34-Res18|DKD|71.70|2.80|1.95|73.07|1.61|1.16|
> > > |Res34-Res18|DIST|72.07|3.33|2.32|73.52|2.24|1.61|
> > > |Res34-Res18|vanilla KD|70.66|1.30|0.91|73.46|2.16|1.55|
> > > |Res50-MBV2|Scratch|68.87|-|-|**72.95**|-|-|
> > > |Res50-MBV2|DKD|72.05|4.62|3.18|73.71|1.04|0.76|
> > > |Res50-MBV2|DIST|73.24|6.35|4.37|74.26|1.8|1.31|
> > > |Res50-MBV2|vanilla KD|68.58|-0.42| -0.29 |74.23|1.75|1.28|
> > >
> > > on CIFAR-100:
> > > |Teacher-Student|Method|previous recipe|effective gain|direct gain|stronger recipe|effective gain|direct gain|
> > > |:-:|:-:|:-:|:-:|:-:|:-:|:-:|:-:|
> > > |Res56-Res20|From Scratch|69.09|-|-|**71.83**|-|-|
> > > |Res56-Res20|DKD|71.97|4.17|2.88|73.10|1.77|1.27|
> > > |Res56-Res20|DIST|71.78|3.89|2.69|74.51|3.73|2.68|
> > > |Res56-Res20|vanilla KD|70.66|2.27|1.57|72.34|0.71|0.51|
> > > |Res32x4-Res8x4|Scratch|72.50|-|-|**74.95**|-|-|
> > > |Res32x4-Res8x4|DKD|76.32|5.27|3.82|78.15|4.27|3.2|
> > > |Res32x4-Res8x4|DIST|75.79|4.54|3.29|77.69|3.66|2.74|
> > > |Res32x4-Res8x4|vanilla KD|73.33|1.14|0.83|75.90|1.27|0.95|

---

> > > > ### Author Response · Authors · 2023-08-15
> > > > **Response to Reviewer aER7 part (2/2)**
> > > >
> > > > **Response to Q1**: The experiments on object detection tasks can be found in the **response to Reviewer bACv**.
> > > >
> > > > **Response to Q2**: As shown in the table above, the "effective gain" is defined as $\frac{acc(distilled)−acc(student)}{acc(student)}$, and "direct gain" is $acc(distilled)−acc(student)$, similar to the defination in LKBT [1] (we changed the denominator to $acc(student)$).
> > > >
> > > > [1] Are Large Kernels Better Teachers than Transformers for ConvNets. ICML 2023
> > > >
> > > > Regarding the results, we observe that in comparison to the improvements seen under the ordinary training recipe, the improvements resulting from KD diminish when the training recipe is strengthened. Nevertheless, the approximately 2% improvement on ImageNet is still substantial, highlighting that the gap between vanilla KD and other KD methods is not as significant as initially perceived.

---

> > > ### Author Response · Authors · 2023-08-17
> > > **Official Comment by Authors**
> > >
> > > Dear reviewer aER7:
> > >
> > > We sincerely thank you for the review and comments.
> > >
> > > We have provided corresponding responses and results, which we've tried our best to cover your concerns. Please let us know whether your concerns have been well addressed. We would like to further discuss with you if you still have any unclear parts of our work.
> > >
> > > Best,
> > >
> > > The Authors

---

> > > > ### Comment · Reviewer_aER7 · 2023-08-17
> > > >
> > > > Thanks for the later response.
> > > >
> > > > According to the results, DKD or DIST is still a better choice for classification and some downstream tasks under an ordinary training recipe. Only under some especially strong training settings, KD performs much better. In this way, the conclusion of this paper does not bring much value to the community actually. Thus I keep my score.

---

> > > > > ### Author Response · Authors · 2023-08-17
> > > > > **Response to Reviewer aER7**
> > > > >
> > > > > Dear Reviewer aER7,
> > > > >
> > > > > Thank you for your valuable feedback. We would like to clarify two points. Firstly, we continuously acknowledge that DKD or DIST remains a better choice under an ordinary training recipe, referring to the conventional practice of training on ImageNet for only 90 epochs. Nevertheless, our training adjustments do not entail an "especially strong training setting," but rather reflect training strategies akin to those outlined in [1,2,3]. Secondly and more importantly, we aim to demonstrate to the community the limitations of exclusively concentrating on KD approaches solely on small-scale datasets. We assert that this practice might not offer a comprehensive understanding of real-world scenarios, encompassing practical usage beyond the confines of academic protocols.
> > > > >
> > > > > [1] Wightman, Ross, Hugo Touvron, and Hervé Jégou. "Resnet strikes back: An improved training procedure in timm." arXiv preprint arXiv:2110.00476 (2021).
> > > > >
> > > > > [2] Touvron, Hugo, et al. "Training data-efficient image transformers & distillation through attention." International conference on machine learning. PMLR, 2021.
> > > > >
> > > > > [3] Huang, Tao, et al. "Knowledge distillation from a stronger teacher." Advances in Neural Information Processing Systems 35 (2022): 33716-33727.

---

### Official Review · Reviewer_bACv · 2023-06-19

**Soundness:** 3 good
**Presentation:** 3 good
**Contribution:** 3 good
**Rating:** 6
**Confidence:** 5

**Summary:**

The paper explores the effectiveness of knowledge distillation (KD) approaches for limited-capacity architectures based on small-scale datasets. The authors identify the "small data pitfall" in previous KD methods, which leads to underestimation of the power of the vanilla KD framework on large-scale datasets like ImageNet-1K.

**Strengths:**

1. Although this work does not propose a novel distillation method, the identification of small data pitfall and a series of analyzes based on this would provide valuable insights for the vision community in the field of knowledge distillation.
2. The experiment is very complete and sufficient, with some persuasiveness.
3. The findings of this article are very important and may correct the research direction of KD.

**Weaknesses:**

1. This article mainly focuses on experimental comparison, while neglecting theoretical analysis. Lack of in-depth analysis of the underlying causes of the observations.
2. The author has overlooked a phenomenon where the KD improvement on large-scale datasets (i.e. ImageNet-1K) is inherently small. Why is this? What is the impact of this on the "small data pitfall" proposed in this paper?

**Questions:**

1. For classification tasks, if that's still the case, what about other tasks?
2. Is there another possibility that this phenomenon is not solely due to the size of the data, but rather to the difficulty or performance bottleneck of the classification task？
3. Can you provide a presentation of the classification results on a certain dataset, and compare the differences between DIST and vanilla KD in terms of whether the classification of certain key samples is correct or not? What about using a subset of imagenet-1k for training and distillation?

---

> ### Author Rebuttal · Authors · 2023-08-10
>
> # Response to Reviewer bACv part (1/2)
>
> **Response to weakness 1:** Thanks for your valuable comments. Our observation suggests that, in the context of knowledge distillation tasks, smaller training sets exhibit a preference for methods featuring stronger priors. These methods effectively impart more informative knowledge [1,2,3] to the student, compensating for the limited data available. Conversely, larger training sets negate the need for such measures. The substantial volume of diverse data inherent to larger datasets naturally imparts ample information for the student model's learning. In such scenarios, a simpler knowledge distillation approach might be adequate, as the teacher model is apt at capturing the inherent complexities within the data, obviating the requirement for extensive priors. In addition, when comparing subsets and the complete ImageNet-1K dataset, the discrepancy between vanilla KD and other methods narrows as the training set scale increases. This observation leads us to believe that informative 'dark' knowledge can naturally manifest within larger datasets. Similarly, Stanton et al. [4] evaluates the fidelity of KD and points out that the student fidelity will increase as the dataset grows: "Enlarging the distillation dataset beyond the teacher training data makes it easier for the student to identify the correct solution, but also makes an already difficult optimization problem harder."
>
> We have conducted additional experiments in our response to ***weakness 1 of mQmQ***, and we paste the results in the table below. To verify our analysis, we show the average confidence on all samples and the entropy of predictions of pretrained Res50/Res152 teacher models on CIFAR-100/ImageNet-1K. We use the two following measurements:
>
> $
> \max({p}(x))=\mathbb{E}_{x\in\mathcal{X}}[\max_i(p_i)],
> $
>
> and
>
> $
> \text{entropy}({p}(x))=\mathbb{E}_{x\in\mathcal{X}}[\textstyle \sum_i p_i\log p_i],
> $
>
> where $p_i$ is predictive probability of sample $x$ belonging to class $i$.
>
> The average confidence score provides an insight into how certain the teacher's predictions are across all samples. As shown in the table, model trained on CIFAR-100 exhibits higher average confidence and lower entropy value, suggesting that the model is more confident in its predictions, however, resulting in reduced mutual information or knowledge transfer among other classes.
>
> ||Top-1|$\max({p}(x))$|$\text{entropy}({p}(x))$|
> |:-:|:-:|:-:|:-:|
> |CIFAR-100|79.33|89.78|0.3362|
> |ImageNet-1K|75.81|79.63|0.7593|
>
> We will add these analyses to our next version.
>
> [1] Huang, Tao, et al. "Knowledge distillation from a stronger teacher." Advances in Neural Information Processing Systems 35 (2022): 33716-33727.
>
> [2] Zhao, Borui, et al. "Decoupled knowledge distillation." Proceedings of the IEEE/CVF Conference on computer vision and pattern recognition. 2022.
>
> [3] Yang, Zhendong, et al. "Rethinking knowledge distillation via cross-entropy." arXiv preprint arXiv:2208.10139 (2022).
>
> [4] Stanton, Samuel, et al. "Does knowledge distillation really work?." Advances in Neural Information Processing Systems 34 (2021): 6906-6919.
>
> **Response to weakness 2:** If I understand correctly, one example of the phenomenon you've mentioned is that most KD approaches can only bring about 1% improvement on top-1 accuracy on ImageNet-1K, but can get about 3-5% improvement on CIFAR-100. Please correct me if I am mistaken. The reasons behind such phenomenon are two-fold. (1) CIFAR-100 is a small-scale dataset consisting of 50K/10K training/testing samples. It is hard for a model trained on such a small dataset to capture the real distribution well. Therefore, the improvement on the student's performance is significant by using the teacher's predictions as additional supervision. However, the ImageNet-1K dataset consists of 1.2M samples, where the diverse data itself is sufficient for the student model to learn the distribution well enough. In such case, the introduction of teacher model only help improve the student performance marginally. (2) Existing approaches usually use small networks such as Res18 and WRN-16 on CIFAR-100, while they use large networks such as Res50 on ImageNet. The difficulty of optimizing models of different sizes (the number of parameters) varies. Some studies [1] indicated that different optimization approaches also influence the results. The previously widely used 90-epoch protocol on ImageNet might now be considered insufficient. This is also aligned with the outcomes from our 'stronger recipe' in Table 1. It seems that we need to use a larger scale dataset to more comprehensively assess the performance of models and KD methods.
>
> [1] Ross Wightman, Hugo Touvron, and Hervé Jégou. Resnet strikes back: An improved training procedure in 420 timm. arXiv preprint arXiv:2110.00476, 2021.
>
> **Response to question 1:** Thanks for your valuable comments. We conduct experiments on COCO (118K training images) object detection task to directly compare DKD, DIST, and vanilla KD. Adhering to DKD's implementation, we applied distillation to the classification branch, with ResNet101 serving as the teacher backbone and ResNet18 as the student backbone. The outcomes indicate that vanilla KD and DIST achieve comparable performance, while DKD surpasses the other two methods. If we consider our assumption of the 'small data pitfall,' we conjecture that this result could be attributed to the limited scale of the COCO dataset. In the future, we plan to validate this speculation on a larger dataset, Objects365, comprising 1720K training images.
>
> |Iteration|Method|AP|AP50|AP75|
> |:-:|:-:|:-:|:-:|:-:|
> |180K|FromScratch|33.26|53.61|35.26|
> |180K|DKD|35.07|56.32|37.45|
> |180K|DIST|34.52|55.64|37.33|
> |180K|KD|34.75|55.95|37.31|
> |540K|DKD|37.44|58.75|40.37|
> |540K|DIST|37.01|58.00|40.07|
> |540K|KD|36.99|57.92|39.87|
>
> ### ***The part (2/2) can be found in "Author Rebuttal" at the top of this page.***

---

> > ### Comment · Reviewer_bACv · 2023-08-13
> > **Response to rebuttal**
> >
> > Thanks for your rebuttal. There are still some concerns. The concept of  "small data pitfall" is not clear, e.g. the scale of datasets, what is small or large? CIFAR is small vs. imagenet is large, imagenet is small vs. SAM-11M/LAION-400M is large ?
> > The experiments on COCO express nothing valuable, just comparsion of some KD methods. COCO is small or large ? The KD methods improve the AP on COCO.

---

> > > ### Author Response · Authors · 2023-08-13
> > > **Response to Reviewer bACv part (1/2)**
> > >
> > > Thank you for your valuable comments.
> > >
> > > We acknowledge that we didn't sufficiently clarify the definition of "small" in our paper. The "small data pitfall" we observed in our main paper refers to the scenarios where the conclusions drawn from certain experiments on datasets which have not reached a certain threshold might not hold in larger dataset contexts which have surpassed the threshold, such as ImageNet and LAION. Our experiments have now narrowed down this threshold of the dataset size to 60% the size of ImageNet, which is shown in Table14 of the supplementary material. This means that **different conclusions may emerge** once the **dataset size exceeds the threshold** (Number of training samples > 0.7M). For example, DIST shows no significant improvements over the baseline vanilla KD. One of the key insights we aim to convey in our paper is that exclusively focusing on KD approaches with small-scale datasets may limit our comprehensive understanding in real-world scenarios. Admittedly, scaling up from ImageNet to even larger datasets, such as LAION, could introduce deviations in existing conclusions. However, our current computational resources may not allow us to perform experiments on such larger datasets. We will update the original paper to provide a clearer description of "small" and "large", emphasizing that once a sufficient quantity of training data is reached, vanilla KD's performance can be similar to that of carefully designed KD methods. To be specific, we will refine the original Line 38-45 to "We point out the small data pitfall in current knowledge distillation literature: once a sufficient quantity of training data is reached, different conclusions emerge. For example,  when evaluated on CIFAR-100 (50K training images), KD methods meticulously designed on such datasets can easily surpass vanilla KD. However, when evaluated on datasets with larger scale, i.e., ImageNet-1K (1M training images), vanilla KD achieves on par or even better results compared to other methods."
> > >
> > > To further investigate the impact of data on knowledge distillation in the context of detection tasks, we are conducting additional ablation studies on the Pascal VOC and COCO datasets. Specifically, the Pascal VOC dataset consists of 20 object classes. Our training set is the combination of VOC 2007 trainval (5K) and VOC 2012 trainval (11K), with the validation set being VOC 2007 test (4.9K), following previous protocals. The training samples in Pascal VOC (16K) are smaller than those in COCO (118K). Therefore, we will also utilize subsets of COCO (30% and 60%) for conducting experiments.
> > >
> > > |Iteration|Method|Teacher-Student-Dataset|mAP|effective gain|direct gain|AP50|AP75|
> > > |:-:|:-:|:-:|:-:|:-:|:-:|---|---|
> > > |18K|From Scratch|N/A-Res18-VOC|42.88|-|-|72.79|43.82|
> > > |18K|DKD|Res101-Res18-VOC|48.07|12.10|5.19|78.75|50.16|
> > > |18K|DIST|Res101-Res18-VOC|47.64|11.10|4.76|78.28|50.01|
> > > |18K|KD|Res101-Res18-VOC| 46.95 |     9.49     |    4.07     | 77.42 | 48.62 |
> > >
> > > |Iteration|Method|Teacher-Student-Dataset|mAP|effective gain|direct gain|AP50|AP75|
> > > |:-:|:-:|:-:|:-:|:-:|:-:|:-:|:-:|
> > > |180K|From Scratch|N/A-Res18-30%COCO|27.32|-|-|46.05|28.19|
> > > |180K|DKD|Res101-Res18-30%COCO|31.27|14.46|3.95|51.94|32.49|
> > > |180K|DIST|Res101-Res18-30%COCO|31.67|15.92|4.35|52.28|33.25|
> > > |180K|KD|Res101-Res18-30%COCO|30.32|10.98|3.00|50.51|31.61|
> > >
> > > |Iteration|Method|Teacher-Student-Dataset|mAP|effective gain|direct gain|AP50|AP75|
> > > |:-:|:-:|:-:|:-:|:-:|:-:|---|---|
> > > |180K|From Scratch|N/A-Res18-60%COCO|30.97|-|-|50.78|32.80|
> > > |180K|DKD|Res101-Res18-60%COCO|33.40|7.85|2.43|54.48|35.81|
> > > |180K|DIST|Res101-Res18-60%COCO|33.18|7.14|2.21|54.01|35.46|
> > > |180K|KD|Res101-Res18-60%COCO|32.98|6.49|2.01|53.43|34.92|
> > >
> > > |Iteration|Method|Teacher-Student-Dataset|mAP|effective gain|direct gain|AP50|AP75|
> > > |:-:|:-:|:-:|:-:|:-:|:-:|---|---|
> > > |180K|From Scratch|N/A-Res18-COCO|33.26|-|-|53.61|35.26|
> > > |180K|DKD|Res101-Res18-COCO|35.07|5.44|1.81|56.32|37.46|
> > > |180K|DIST|Res101-Res18-COCO|34.52|3.79|1.26|55.64|37.33|
> > > |180K|KD|Res101-Res18-COCO|34.75|4.48|1.49|55.95|37.31|
> > >
> > > ~~We will **update the table above once the experiments are completed**, and provide corresponding analyses for the object detection task.~~ While Objects365 is large enough and also available for object detection, we are concerned that we may lack sufficient computational resources to conduct experiments on it. Our primary focus remains on the more widely applied field of KD in classification, which is more extensively used at present.

---

> > > > ### Author Response · Authors · 2023-08-15
> > > > **Response to Reviewer bACv part (2/2)**
> > > >
> > > > As shown in the table above, the "effective gain" is defined as $\frac{acc(distilled)−acc(student)}{acc(student)}$, and "direct gain" is $acc(distilled)−acc(student)$, similar to the defination in LKBT [1] (we changed the denominator to $acc(student)$ to better accommodate ablations with subsets).
> > > >
> > > > [1] Are Large Kernels Better Teachers than Transformers for ConvNets. ICML 2023
> > > >
> > > > The experiments on detection tasks cover four datasets: (i) VOC, with 16K training samples and 20 categories; (ii) 30% COCO, with 35K training samples and 80 categories; (iii) 60% COCO, with 71K training samples and 80 categories; and (iv) full COCO, with 118K training samples and 80 categories. From the experimental results, it can be observed that on datasets with similar difficulty levels, the performance gap between vanilla KD and DKD gradually diminishes as the number of training samples increases. For example, on the 30% COCO dataset, the direct gain for vanilla KD compared to DKD is 0.95 (effective gain of 3.48), while on the full COCO dataset, it is 0.32 direct gain (effective gain of 0.96). **This phenomenon is analogous to the "small data pitfall" observed in the classification task, where as the dataset size grows, the performance gap between vanilla KD and other KD methods narrows.** Additionally, in both the VOC and 30% COCO settings (the use of relatively small training datasets), despite different task difficulties (categories), the performance gap between vanilla KD and DKD methods remains similar, with a 1.12 direct gain (effective gain of 2.61) vs. 0.95 direct gain (effective gain of 3.48).

---

> > > > ### Comment · Reviewer_bACv · 2023-08-20
> > > > **Thanks for response**
> > > >
> > > > Thank you for the response. My concerns have been addressed. Therefore, I'd like to increase the score.

---

> > > > > ### Author Response · Authors · 2023-08-20
> > > > >
> > > > > Thank you for your time and dedication. Your feedback on our rebuttal would be greatly appreciated.

---

> > > ### Author Response · Authors · 2023-08-17
> > > **Official Comment by Authors**
> > >
> > > Dear reviewer bACv:
> > >
> > > We sincerely thank you for the review and comments.
> > >
> > > Please let us know whether your concerns have been well addressed. We would like to further discuss with you if you still have any unclear parts of our work.
> > >
> > > Best,
> > >
> > > The Authors

---

> > > ### Author Response · Authors · 2023-08-19
> > >
> > > Dear reviewer bACv,
> > >
> > > We greatly appreciate your insightful comments and suggestions. As the discussion phase draws to a close, we are reaching out to inquire if you have any further feedback on our response. We are open and receptive to any additional queries or concerns you might have. Your feedback and ongoing engagement are of immense value to us.
> > >
> > > In this paper, we delve into a critical question within the KD community: whether previous approaches remain effective in more intricate scenarios involving larger datasets. Our exploration on conventional classification task indicates that the conclusions drawn from datasets that haven't reached a specific threshold (e.g., CIFAR) might not apply to larger dataset contexts that have exceeded the threshold (e.g., ImageNet). We term this phenomenon as the "small data pitfall". Through our experiments, we've now pinpointed this dataset size threshold to be around 60% of the ImageNet size, as evident in Table 14 of the supplementary material. This implies that once the dataset size exceeds this threshold (with over 0.7M training samples), different conclusions might emerge. For example, in the ImageNet classification task, DIST shows no significant improvements over the vanilla KD baseline. A key insight we aim to convey in our paper is that a sole focus on KD approaches using small-scale datasets could limit our comprehensive grasp of real-world scenarios. Furthermore, our additional experiments on detection tasks illustrate that for datasets with comparable difficulty levels, the performance gap between vanilla KD and DKD gradually diminishes as the number of training samples increases. This observation mirrors the "small data pitfall" noted in the classification task, where, as the dataset size grows, the performance gap between vanilla KD and other KD methods narrows. Additionally, we leverage vanilla KD to elevate the performance of various architectures like ResNet-50, ViT-Tiny, ViT-Small, and ConvNeXtV2 beyond their previously reported best results.
> > >
> > > Best,
> > >
> > > The Authors

---

### Official Review · Reviewer_mQmQ · 2023-07-04

**Soundness:** 4 excellent
**Presentation:** 4 excellent
**Contribution:** 3 good
**Rating:** 8
**Confidence:** 5

**Summary:**

This paper revisits vanilla knowledge distillation and presents an empirical analysis of the impact of model size, dataset scale, and training strategy on student performance in knowledge distillation. It identifies: 1) the gap between vanilla KD and other carefully designed KD methods gradually diminishes when adopting stronger data augmentation techniques and longer training iterations on large-scale datasets such as ImageNet-1K; 2) on small scale datasets, vanilla KD consistently underperforms other KD approaches although stronger training strategy and longer training iterations are used; and 3) logits-based methods outperform hint-based methods in terms of generalizability in more challenging cases. Based on these observations, this paper trains four different student models achieving SOTA performance on ImageNet-1K solely using the vanilla KD method.

**Strengths:**

1. Personally, I agree that vanilla KD can serve as a formidable contender when employed with large-scale datasets. Hints serve as valuable priors during training **only** when the available data is severely limited.
2. The authors diligently conducted extensive experiments to thoroughly examine the performance of vanilla KD across various factors, such as model capacity, dataset size, training epochs, learning rate, and regularization techniques, among others.
3. Vanilla KD is indeed a practical approach with widespread applications, making this paper highly beneficial for a diverse range of readers.
4. Notably, this paper presents state-of-the-art (SOTA) models trained using vanilla KD, which can significantly aid in further research.

**Weaknesses:**

- The evaluation of vanilla KD in this work is solid and self-contained. When dealing with a small training set, it is customary to incorporate additional priors, regularization techniques, augmentations, or hints like previous KD methods. Naturally, with a larger dataset, we can leverage vanilla KD with less regularisation to attain comparable performance levels. However, it would be better if the authors can provide a systematical analysis for the choice of algorithms under Small Scale & Large Scale settings.
- The paper compares teacher models in two aspects: model size and dataset scale. Adopting a teacher model with more parameters is beneficial (ResNet152 vs. BEiTv2-L), but an extremely large teacher is harmful (BEiTv2-B vs. BEiTv2-L). As for the dataset scale, training the teacher on larger datasets is helpful. I wonder if we continue to increase the dataset scale, whether it will show a trend similar to increasing the model size, i.e., the student performance decreases. If not, what leads to this difference?
- The conclusion of comparison KD with MIM is a bit weak as it assumes a stronger teacher model is available. In practice, this may not always hold. For example, if we want to train a BeITv2-L, perhaps BeITv2-L is not a good teacher, and we cannot find one achieving higher accuracy. So I think KD is only preferable when training small- or medium-size models.

Minor typos:
1. "reflection on whether" in line 73
2. "results show that" in line 127
3. "experimental setup is as follows" in line 136
4. "other approaches" in line 200
5. "that trained" in line 243
6. "learn from" in line 304


**Questions:**

Please refer to the weaknesses.

---

> ### Author Rebuttal · Authors · 2023-08-10
>
> **Response to weakness 1:** Thanks for your valuable suggestions. The trend is that a small training set prefers a knowledge distillation (KD) method with stronger regularization or priors, as it can effectively bring in more informative knowledge from the teacher model to enhance the performance of the smaller student model. On the other hand, with a large training set, the need for additional prior diminishes. The sheer abundance of diverse data in a larger dataset already provides sufficient information for the student model to learn from. In such cases, a simpler KD approach may suffice, as the teacher model can effectively capture the complexities present in the data without requiring extensive regularization or priors.
>
> To verify the above analysis, we present the average confidence on all samples and the entropy of predictions of pretrained ResNet50/ResNet152 teacher models on CIFAR-100/ImageNet-1K. We use the two following measurements:
>
> $
> \max({p}(x))=\mathbb{E}_{x\in\mathcal{X}}[\max_i(p_i)],
> $
>
> and
>
> $
> \text{entropy}({p}(x))=\mathbb{E}_{x\in\mathcal{X}}[\textstyle \sum_i p_i\log p_i],
> $
>
> where $p_i$ is predictive probability of sample $x$ belonging to class $i$.
>
> The average confidence score provides an insight into how certain the teacher's predictions are across all samples. As shown in the table, model trained on CIFAR-100 exhibits higher average confidence and lower entropy value, suggesting that the model is more confident in its predictions, however, resulting in reduced mutual information or knowledge transfer among other classes.
>
> ||Top-1|$\max({p}(x))$|$\text{entropy}({p}(x))$
> |:-:|:-:|:-:|:-:
> |CIFAR-100|79.33|89.78|0.3362
> |ImageNet-1K|75.81|79.63|0.7593
>
> > **Weakness 2:** increase the dataset scale
>
> To demonstrate this, we first analyze the impact of model size by comparing the average confidence on all samples and the entropy of predictions of ResNet152/BEiTv2-B/BEiTv2-L teacher model on ImageNet-1K.
>
> ||Top-1|$\max({p}(x))$|$\text{entopy}({p}(x))$|
> |:-:|:-:|:-:|:-:|
> |ResNet152|82.83|67.46|2.4418|
> |BEiTv2-B|86.39|80.71|1.2868|
> |BEiTv2-L|88.39|83.94|1.1172|
>
> As depicted in the table, as the teacher model size increases, their predictions become less informative. Our hypothesis is that smaller teacher models may struggle to learn the dataset adequately, leading to noisy soft labels that are less beneficial for the student. On the other hand, larger teacher models tend to overfit the dataset, hindering the distillation process with their soft labels (more closely resembling one-hot ground truth and less transferable knowledge to other classes). Consequently, the model of medium size, BEiTv2-B, exhibits the best performance in our experiments
>
> Based on the analysis above, teacher models tend to overfit smaller datasets, leading to a lack of valuable information for distillation. Recent KD approaches have outperformed vanilla KD by introducing additional information on small-scale datasets. However, as the dataset scale increases, the overfitting issue diminishes, making it challenging to optimize the objectives of complex KD methods due to an abundance of information for the student to learn. On the contrary, the simplicity of vanilla KD, which solely relies on soft labels, becomes sufficient for distillation.
>
> Consequently, we expect an improvement in the student's performance owing to the ample information provided by the increasing dataset scale. And we believe that as the dataset scale further increases, vanilla KD will achieve comparable performance to other KD baselines. The reduction in overfitting with larger datasets mitigates the need for more intricate KD approaches, reaffirming the effectiveness of vanilla KD for knowledge transfer in such scenarios.
>
> **Response to weakness 3:** Indeed, there are instances where a more powerful teacher is not accessible. To assess the effectiveness of vanilla KD under such circumstances, we conduct experiments on ImageNet-1K, employing the identical architecture for both the teacher and student models. In this regard, we utilize the BEiTv2-B model as our chosen architecture and proceed to compare the performance of vanilla KD against the MIM pretraining method introduced by BEiTv2 [1], evaluating accuracy and training time consumption as key metrics. The GPU time measurements are conducted using a single NVIDIA Tesla V100 GPU.
>
> |Method|Epoch|Top-1|GPUtime(hours)
> |:-:|:-:|:-:|:-:
> |MIM|300(pretrain)+100(finetune)|85.0|853
> |MIM|1600(pretrain)+100(finetune)|85.5|3979
> |vanillaKD|300|84.6|575
> |vanillaKD|600|85.4|1151
> |vanillaKD|1600|85.7|3069
>
> Based on above results, it's evident that the performance of vanilla KD is on par with that of MIM. For example, the student model trained through vanilla KD (600 epochs) attains a top-1 accuracy of 85.4%, only 0.1% lower than the equivalent model trained with MIM (1600 epochs). However, the vanilla KD approach requires notably less training time. Furthermore, extending the training epochs to 1600 yields a top-1 accuracy of 85.7%, surpassing the MIM result by 0.2% under the same epoch settings. This underscores that vanilla KD remains competitive with MIM, even when a more robust teacher model is unavailable.
>
> Specifically, we choose the BEiTv2-B with 85.5% top-1 accuracy (the 2nd line) as the teacher model in KD. In our reported GPU time, we omitted the training duration of the teacher model. Given that it is typically feasible to procure a pre-trained model with performance comparable to the student model, the necessity to train a teacher model from the ground up is minimized in most scenarios.
>
> [1] Beit v2: Masked image modeling with vector-quantized visual tokenizers
>
> **Response to weakness 4:** Thank you for thoroughly reviewing our submission and pointing out the mistakes. We will take great care to address these issues in our next version. Your feedback is invaluable in improving the quality of our work.

---

> > ### Comment · Reviewer_mQmQ · 2023-08-13
> > **Replying to Rebuttal**
> >
> > Thank you for the feedback. I have thoroughly read the rebuttal and comments from the other reviewers. I don't have any additional questions and will maintain the initial score.

---

> > > ### Author Response · Authors · 2023-08-13
> > > **Response to Reviewer mQmQ**
> > >
> > > Dear Reviewer mQmQ,
> > >
> > > We sincerely appreciate you taking time to review our paper and response. We will carefully follow reviewer's advice to incorporate the addressed points in updated version.
> > >
> > > Best,
> > > Authors of Paper 1852

---

### Official Review · Reviewer_oNYG · 2023-07-04

**Soundness:** 2 fair
**Presentation:** 3 good
**Contribution:** 3 good
**Rating:** 6
**Confidence:** 3

**Summary:**

This paper investigates the effectiveness of vanilla Knowledge Distillation (KD) in large-scale datasets. The authors identify a "small data pitfall"  which underestimates the power of vanilla KD on large-scale datasets and demonstrate that stronger data augmentation techniques and larger datasets can decrease the gap between vanilla KD and other KD variants.

**Strengths:**

- **Originality**: This is an empirical study. It has no technical novelty but sheds some insights into the knowledge distillation method.

- **Quality**: Extensive experiments validate their conclusions.

- **Clarity**: This paper is well-structured and easy to follow.

- **Significance**: Knowledge Distillation is an important and interesting topic and exploring the impacts of the size of datasets for KD is beneficial for the community.

- New state-of-the-art results for ResNet-50, ConvNeXt-T architectures.

**Weaknesses:**

-  Lightweight models, such as MobileNetv3, are deemed critical model architectures due to their efficiency and compactness, which make them ideal for deployment on devices with limited computational resources. However, it appears that there is a lack of dedicated experiments specifically for these lightweight models.

- Two related papers are missing[1][2]

[1] Yang, Zhendong, et al. "Rethinking knowledge distillation via cross-entropy." arXiv preprint arXiv:2208.10139 (2022).

[2] Huang, Tianjin, et al. "Are Large Kernels Better Teachers than Transformers for ConvNets?." ICML. 2023.





**Questions:**

- Temperature is an important hyper-parameter for the KD method. what value is used for temperature in all experiments? How does the temperature affect the vanilla KD performance from small to large datasets?

- Does the conclusion still hold in lightweight models such as MobileNetv3?

**Limitations:**

 the authors adequately addressed the limitations

---

> ### Author Rebuttal · Authors · 2023-08-10
>
> > **Weakness 1:** Lightweight models, such as MobileNetv3, are deemed critical model architectures due to their efficiency and compactness, which make them ideal for deployment on devices with limited computational resources. However, it appears that there is a lack of dedicated experiments specifically for these lightweight models.
> >
> > **Question 2:** Does the conclusion still hold in lightweight models such as MobileNetv3?
>
> **Response to weakness 1 & question 2:** Thank you for your valuable suggestion. We have opted to employ MobileNetv3 as a representative lightweight model for our experimental investigations, as shown in the following table. The outcomes reveal that on the CIFAR-100 dataset, vanilla KD falls short of the leading baseline, DKD, by 0.51% in terms of top-1 accuracy. Nevertheless, on the ImageNet-1K dataset, vanilla KD emerges as the top performer. These results reaffirm the validity of the small data pitfall observed in our main paper.
>
> |   Dataset   |  Teacher   |      Student       | Method |   Top-1   |
> | :---------: | :--------: | :----------------: | :----: | :-------: |
> |  CIFAR-100  |     -      | MobileNet v3 Small |   -    |   64.76   |
> |  CIFAR-100  | ResNet32x4 | MobileNet v3 Small |  DKD   | **69.10** |
> |  CIFAR-100  | ResNet32x4 | MobileNet v3 Small |  DIST  |   67.72   |
> |  CIFAR-100  | ResNet32x4 | MobileNet v3 Small |   KD   |   68.59   |
> | ImageNet-1K |     -      | MobileNet v3 Small |   -    |   67.40   |
> | ImageNet-1K |  BeiTv2-B  | MobileNet v3 Small |  DKD   |   67.36   |
> | ImageNet-1K |  BeiTv2-B  | MobileNet v3 Small |  DIST  |   68.02   |
> | ImageNet-1K |  BeiTv2-B  | MobileNet v3 Small |   KD   | **68.05** |
>
> > **Weakness 2:** Two related papers are missing
>
> **Response to weakness 2:** Thank you for bringing the overlooked related papers to our attention. [1] decomposes the original KD loss into two components and proposes NKD loss to discard the challenging-to-optimize portion. [2] points out that large-kernel ConvNets are more effective teachers than Transformers for small-kernel ConvNets. We will incorporate references to these two papers in the related work section of our next version.
>
> > **Question 1:** Temperature is an important hyper-parameter for the KD method. what value is used for temperature in all experiments? How does the temperature affect the vanilla KD performance from small to large datasets?
>
> **Response to question 1:** In our experiments, we follow the implementation of DKD to employ the temperature setting of T=4 for all experiments on CIFAR-100 and T=1 for those on ImageNet-1K.
>
> To assess the influence of temperature in knowledge distillation (KD), we conducted experiments on both the CIFAR-100 and ImageNet-1K datasets. As indicated in the table below, the effect of temperature on student performance is non-monotonic. However, a discernible trend emerges: larger temperature values (T > 1) tend to be more effective for small-scale datasets, while a smaller temperature value (T=1) is preferable for larger-scale datasets. We posit that smaller-scale datasets often lead to teacher models overfitting, resulting in predictions with low entropy. Consequently, softening these predictions with a higher temperature can yield more valuable information for the student. Conversely, in the context of larger-scale datasets, teachers inherently provide less certain predictions, and elevating uncertainty further by using a higher temperature setting could hinder student learning.
>
> |   Dataset   |  Teacher   |      Student       |  Method  | Top-1 |
> | :---------: | :--------: | :----------------: | :------: | :---: |
> |  CIFAR-100  | ResNet32x4 | MobileNet v3 Small | KD (T=1) | 67.07 |
> |  CIFAR-100  | ResNet32x4 | MobileNet v3 Small | KD (T=2) | 69.35 |
> |  CIFAR-100  | ResNet32x4 | MobileNet v3 Small | KD (T=4) | 68.59 |
> | ImageNet-1K |  BeiTv2-B  |      ResNet50      | KD (T=1) | 80.96 |
> | ImageNet-1K |  BeiTv2-B  |      ResNet50      | KD (T=2) | 80.39 |
> | ImageNet-1K |  BeiTv2-B  |      ResNet50      | KD (T=4) | 80.53 |
>
> We have also added experiments concerning the temperature settings for DKD and DIST on ImageNet-1K. The specific results are provided below. We will incorporate these findings to Table 16 in our supplementary material.
>
> |   Dataset   | Teacher  | Student  |   Method   | Top-1 |
> | :---------: | :------: | :------: | :--------: | :---: |
> | ImageNet-1K | BeiTv2-B | ResNet50 | DIST (T=1) | 80.76 |
> | ImageNet-1K | BeiTv2-B | ResNet50 | DIST (T=2) | 80.46 |
> | ImageNet-1K | BeiTv2-B | ResNet50 | DIST (T=4) | 80.31 |
> | ImageNet-1K | BeiTv2-B | ResNet50 | DKD (T=1)  | 80.94 |
> | ImageNet-1K | BeiTv2-B | ResNet50 | DKD (T=2)  | 80.76 |
> | ImageNet-1K | BeiTv2-B | ResNet50 | DKD (T=4)  | 80.70 |

---

> > ### Comment · Reviewer_oNYG · 2023-08-12
> >
> > Thank you for the rebuttal. Most of my concerns have been addressed. Therefore, I increase my score to 6.

---

> > > ### Author Response · Authors · 2023-08-12
> > > **Response to Reviewer oNYG**
> > >
> > > Dear Reviewer oNYG,
> > >
> > > We sincerely appreciate you taking time to review our responses and contributing to improve this paper. We will carefully follow reviewer's advice to incorporate the addressed points in updated version.
> > >
> > > Best,
> > > Authors of Paper 1852

---

### Author Rebuttal · Authors · 2023-08-10

# Response to all reviewers

We thank all four reviewers for their constructive feedbacks which greatly improved the quality of our paper.


### **Response to Reviewer bACv, part (2/2)**

**Response to question 2:** Thanks for the insightful question.  First, we analyzed the impact of task difficulty. In Figure 1 of our main paper, we compared the performance of vanilla KD and two baselines on the complete ImageNet-1K dataset as well as its two subsets. These subsets were obtained through stratified random sampling from the same 1000 classes, thus ensuring that the difficulty level (the number of categories) of tasks remains consistent across the datasets. The results illustrated that as the dataset scale increased, the performance gap between vanilla KD and the baselines diminished. This suggests that the observed phenomenon is driven by the dataset size rather than inherent task difficulty.

Next, we explored the performance bottleneck of the classification task. On the smaller-scale CIFAR-100 dataset, the baselines achieved superior results compared to vanilla KD. This implies that vanilla KD's performance has not yet reached a bottleneck (student can obtain higher accuracy). In the context of the larger-scale ImageNet-1K dataset, our experiments, which utilized an extended training schedule as detailed in Section 3.6, demonstrated that the student's performance did not reach a bottleneck even with training epochs extended to 4800. Since most of our experiments utilized 300/600/1200 epochs, we can confidently discount the performance bottleneck of the classification task as a factor influencing our conclusion.

**Response to question 3:** Thanks for the valuable comments. For the presentation, please refer to our "Author Rebuttal" section where we attached the PDF to show the cases.

As for using a subset of imagenet-1k, we show the results of using a subset of ImageNet-1K, i.e., 30% and 60%, to distill the student model. The results on validation set can be found in Table 14 in our supplementary material. When comparing the outcomes between subsets and the complete ImageNet-1K dataset, the discrepancy between vanilla KD and other methods narrows as the training set scale increases.

We choose the sutdent trained in Figure 1 of our main paper (ResNet50 trained on ImageNet-1K 30%/60%/full) to comapre the differences between DIST and vanilla KD. As the Table 1 shows, students trained by DIST and KD achieve similar performance. However, these two students give different predictions on more than 3000 samples. The result indicates that students trained via vanilla KD and DIST can be be very different at single sample level.

The marker "$\checkmark$" indicates the student trained by the corresponding method gives correct predictions, and marker "$\times$" indicates it gives wrong predictions. The adopted dataset is the validation set of ImageNet-1K, which consists of 50K samples. For example, the first row, DIST ($\checkmark$), indicates that when trained with a subset (30%) of ImageNet, there are 39872 samples correctly classified by DIST. The fourth row, DIST ($\checkmark$) & KD ($\times$), indicates that when trained with a subset (30%) of ImageNet, there are 1904 samples correctly classified by DIST but misclassified by KD.

||subset (30%)|subset (60%)|ImageNet (100%)|
|:-:|:-:|:-:|:-:|
|DIST ($\checkmark$)|39872|40665|40912|
|KD ($\checkmark$)|39781|40668|41137|
|DIST ($\checkmark$) & KD ($\checkmark$)|37968|38982|39270|
|DIST ($\checkmark$) & KD ($\times$)|1904|1683|1642|
|DIST ($\times$) & KD ($\checkmark$)|1813|1686|1867|
|DIST ($\times$) & KD ($\times$)|8315|7649|7221|

We have included visualizations of specific cases in the attached PDF. We selected four categories randomly and displayed the samples that are correctly classified by both DIST and KD, correctly classified by DIST but misclassified by KD, correctly classified by KD but misclassified by DIST, or misclassified by both DIST and KD. Upon analyzing the visualizations, we observed that vanilla KD tends to learn the distribution of the teacher, as seen in cases like the fourth row for the "Binoculars" category where the misclassified class by vanilla KD is the same as the teacher's class. We speculate that this behavior might be attributed to the KL divergence loss, which encourages the student to directly match the teacher's distribution. In contrast, DIST learns more implicit similarity relationships, which could result in a more explicit difference from the teacher's prediction.

### **Response to Reviewer aER7 weakness 4, part (2/2)**

Regarding the temperature parameter, we conducted supplementary experiments to investigate its impact on the performance of knowledge distillation (KD). We conduct experiments on both CIFAR-100 and ImageNet-1K dataset. From the results, the impact of temperature to the student performance is not monotonic, but there is a trend that a small temperature value is better for the small scale dataset while a larger temperature value is better for the larger scale dataset. We speculate that teacher models tend to overfit small scale dataset. Hence, predictions softened by a higher temperature can provide more useful information for the student. However, on large scale dataset, teachers already give uncertain predictions, and further increasing the uncertainty by using a higher temperature setting would impede learning of the student.

|Dataset|Teacher|Student|Method|Top-1|
|:-:|:-:|:-:|:-:|:-:|
|CIFAR-100|ResNet32x4|MobileNet v3 Small|KD(T=1)|67.07|
|CIFAR-100|ResNet32x4|MobileNet v3 Small|KD(T=2)|69.35|
|CIFAR-100|ResNet32x4|MobileNet v3 Small|KD(T=4)|68.59|
|IN-1K|BeiTv2-B|ResNet50|KD(T=1)|80.96|
|IN-1K|BeiTv2-B|ResNet50|KD(T=2)|80.39|
|IN-1K|BeiTv2-B|ResNet50|KD(T=4)|80.53|

---

### Decision · Program_Chairs · 2023-09-21

**Decision:**

Accept (poster)

**Comment:**

This paper explores the potential of Vanilla Knowledge Distillation (KD) approaches for limited-capacity architectures, and identifies the small data pitfall that limits the effectiveness of previous KD methods on large-scale datasets. The main strengths of this paper include its thorough investigation of the crucial factors in deciding distillation performance. The authors demonstrate that using stronger data augmentation techniques and larger datasets can significantly improve the performance of vanilla KD, highlighting the importance of designing and evaluating KD approaches in practical scenarios. The study also compares the effectiveness of vanilla KD to other meticulously designed KD variants, and shows that vanilla KD can achieve state-of-the-art results on ImageNet without the need for complex training strategies or model capacities.

The authors effectively addressed initial concerns raised by reviewers, including new experiments on compact NNs, an ablation study covering regularization, dataset size, and self-distillation, and clarifications regarding small versus large-scale datasets. While there was a debate with Reviewer aER7 about their training recipe (different from conventional ImageNet 90-epoch setting), AC concluded that it followed a reasonable experimental practice in KD literature and should not be penalized.

AC believes that the paper's main contribution is not necessarily in introducing a new SOTA KD method but in revealing the limitations and challenges of evaluating KD on small-scale datasets. The paper effectively conveys this message, making it sound and meaningful, leading AC to recommend its acceptance.